# High proliferation and delamination during skin epidermal stratification

Mareike Damen[1,2,3], Lisa Wirtz [1,2,3], Ekaterina Soroka[1,2], Houda Khatif[1,2], Christian Kukat [4], Benjamin D. Simons [5] & Hisham Bazzi [1,2,6✉]

The development of complex stratified epithelial barriers in mammals is initiated from single-layered epithelia. How stratification is initiated and fueled are still open questions. Previous studies on skin epidermal stratification suggested a central role for perpendicular/asymmetric cell division orientation of the basal keratinocyte progenitors. Here, we use centrosomes, that organize the mitotic spindle, to test whether cell division orientation and stratification are linked. Genetically ablating centrosomes from the developing epidermis leads to the activation of the p53-, 53BP1- and USP28-dependent mitotic surveillance pathway causing a thinner epidermis and hair follicle arrest. The centrosome/p53-double mutant keratinocyte progenitors significantly alter their division orientation in the later stages without majorly affecting epidermal differentiation. Together with time-lapse imaging and tissue growth dynamics measurements, the data suggest that the first and major phase of epidermal development is boosted by high proliferation rates in both basal and suprabasally-committed keratinocytes as well as cell delamination, whereas the second phase maybe uncoupled from the division orientation of the basal progenitors. The data provide insights for tissue homeostasis and hyperproliferative diseases that may recapitulate developmental programs.

[1] Department of Dermatology and Venereology, University Hospital of Cologne, University of Cologne, Cologne, Germany. [2] The Cologne Cluster of Excellence in Cellular Stress Responses in Aging-associated Diseases (CECAD), University of Cologne, Cologne, Germany. [3] Graduate School for Biological Sciences, University of Cologne, Cologne, Germany. [4] FACS & Imaging Core Facility, Max Planck Institute for Biology of Aging, Cologne, Germany. [5] The Welcome Trust/Cancer Research UK Gurdon Institute, University of Cambridge, Cambridge, UK. [6] Center for Molecular Medicine Cologne (CMMC), University of Cologne, Cologne, Germany. ✉email: hisham.bazzi@uk-koeln.de

The generation of complex stratified epithelia, including the skin epidermis, during mammalian embryonic development, is essential to form barriers that are compatible with postnatal life[1]. The stratified skin epidermis is initiated from the single-layered simple epithelium, derived from the ectoderm, surrounding the embryo[2]. Around embryonic day (E) 9.5, the epithelial cells commit to stratification through the master regulator p63, a member of the p53 family of transcription factors[3,4]. The progenitor cells down-regulate the expression of the simple-epithelial keratin intermediate filaments, such as Keratin-8 (K8), and express the complex-epithelial keratins, K5 and K14. The periderm is the first layer generated and it acts as a transient protective and insulating barrier for the developing embryo[5]. Around E12.5, the first differentiated suprabasal epidermal layer cells appear and are characterized by the expression of a distinct set of keratins, K1 and K10[6]. Subsequently, the epidermal keratinocytes undergo further differentiation and crosslinking to generate a fully functional barrier by E17.5[7]. The stereotypical regeneration and differentiation program in the epidermis, starting from the basal stem cells, to replenish the shed corneocytes persists throughout the life of the animal[8]. In addition to the interfollicular epidermal stem cells, the basal keratinocyte progenitors also give rise to all the stem cells of the skin epithelium, including the hair follicle stem cells[9].

The mechanism of how the new layers of the stratified epidermis are generated and maintained is not well-understood. The published data support a model for interfollicular epidermal stratification that is coupled to the orientation of cell division in the basal layer of the epidermis[2,10]. At E12.5 and earlier, almost all of the progenitor basal cells divide with an axis that is parallel to the basement membrane and undergo a presumptive symmetric division to generate two progenitor daughter cells that remain in the basal layer[11]. From E13.5 onwards, more of the dividing basal progenitors shift their axis of division to a perpendicular orientation, which has become synonymous with an asymmetric division, to generate one daughter progenitor cell that remains in the basal layer, and one differentiated daughter cell that stratifies the forming epidermis[11,12]. Several studies in the literature are consistent and correlate with the division orientation-based model[13–19]. For example, manipulations in gene products that result in a thinner epidermis, such as knockdown of LGN or NuMa1, are highly associated with an increase in the fraction of parallel divisions, whereas those causing a thickened hyperdifferentiated epidermis, such as overexpression of INSC, are correlated with an increase in the fraction of perpendicular divisions[12]. Importantly, there is currently no consensus in the field on what stage of epidermal development, for example, E14–E15 versus E16–E17, or phase of mitosis, in particular metaphase versus late anaphase/telophase, should be considered during cell division orientation analyses; however, the majority suggests the later developmental stage as well as late mitotic phase[13–19].

Centrosomes are major microtubule-organizing centers of animal cells that regulate cell division and are composed of a pair of centrioles surrounded by a proteinaceous matrix[20]. Centrosomes are essential to provide the centriolar template for cilia and are important for efficient mitotic spindle assembly[21]. In humans, mutations in genes encoding centrosomal proteins lead to primordial dwarfism and microcephaly[22]. We have previously shown that the constitutive ablation of Sas-4, a gene essential for centriole formation and duplication, leads to the loss of centrioles and cilia in early developing mouse embryos[23]. The loss of centrioles, but not the secondary loss of cilia, results in p53-dependent cell death and embryonic arrest at E9.5[23]. Conditional ablation of Sas-4 in the developing brain recapitulates the human microcephaly phenotype and leads to p53-dependent cell death of the radial glial progenitors (RGPs) in the cortex[24]. Activation of this p53-dependent pathway is independent of DNA damage or chromosome segregation errors and instead is associated with prolonged mitotic duration[23,24]. Recent reports in cultured mammalian cell lines have confirmed our findings and extended them to include 53BP1 and USP28 as components acting upstream of p53 in a pathway now termed "the mitotic surveillance pathway"[25–30].

In this work, we conditionally remove Sas-4/centrioles and p53 from the developing skin epidermis to test whether the keratinocyte basal progenitors can uncouple cell division orientation from epidermal stratification and differentiation. To separate the functions of centrioles in cilia formation versus spindle assembly, we also conditionally remove Ift88, a gene required for the formation of cilia but not centrioles[31–33]. Our data show that the loss of centrioles, but not cilia, result in early p53-dependent cell death, leading to a thinner epidermis and arrested hair follicles. These phenotypes are rescued in the Sas-4 p53 double mutant epidermis, which resembles controls and cilia mutants. Importantly, the double mutant basal keratinocyte progenitors show a significant shift in cell division orientation that is uncoupled from epidermal stratification and differentiation at the later stages of epidermal development. Using time-lapse imaging and measurements of tissue growth dynamics in developing embryos, the data support a two-phase model of epidermal stratification and development that highlights the importance of the early phase (E13–E15) and is based on cell delamination and high proliferation in the basal and suprabasal progenitors.

## Results

**Centrioles are important, but not essential, for mouse epidermal and hair follicle development.** In order to assess the functions of mammalian centrioles and centrosomes in the developing mouse skin epithelium, we deleted Sas-4 using a K14-Cre line[34], which expresses the Cre recombinase in developing stratified epithelia, including the skin epidermis as early as embryonic day (E) 9.5, in combination with a Sas-4 conditional allele[23,24]. Immunostaining for γ-Tubulin (TUBG, a marker for centrosomes) and CEP164 (a marker for centriolar distal appendages) confirmed that centrioles are almost completely lost from the epidermal basal and suprabasal cells by E15.5 (Fig. 1a–c). Consistent with the reduced viability of the centrosome mutant animals by P3 (~60% compared to ~ 95% for control animals), the embryos at E17.5 showed a slight delay in skin barrier formation at the chin and the paws, as judged by a Toluidine Blue dye-penetration assay, which was restored just before birth at E18.5 (Fig. 1d). On a postnatal day 21 (P21), the centrosome mutant mice were smaller than their control littermates and had grossly thin and transparent skin with very sparse hair (Fig. 1e; Supplementary Fig. 1a). Starting as early as P4, the centrosome mutant mice had a significantly reduced weight relative to their littermates (Supplementary Fig. 1a), likely due to the K14-Cre Sas-4 defects in the oral epithelium which impaired food intake. To avoid any secondary complications of centriole loss after birth, we focused most of our analyses on embryonic development until P0.

At P0, the Sas-4 mutant epidermis was significantly thinner than that of control littermates with a marked reduction in the number of hair follicles (Fig. 1e–g). To study if the epidermal differentiation process was affected by the loss of centrioles, we performed immunostainings on back-skin sections at P0. Staining for markers of the proliferative basal layer (K14), and differentiated layers (K1 and Loricrin, LOR), showed that although the epidermis and its differentiated layers were thinner, the differentiation markers were not majorly affected in the Sas-4/

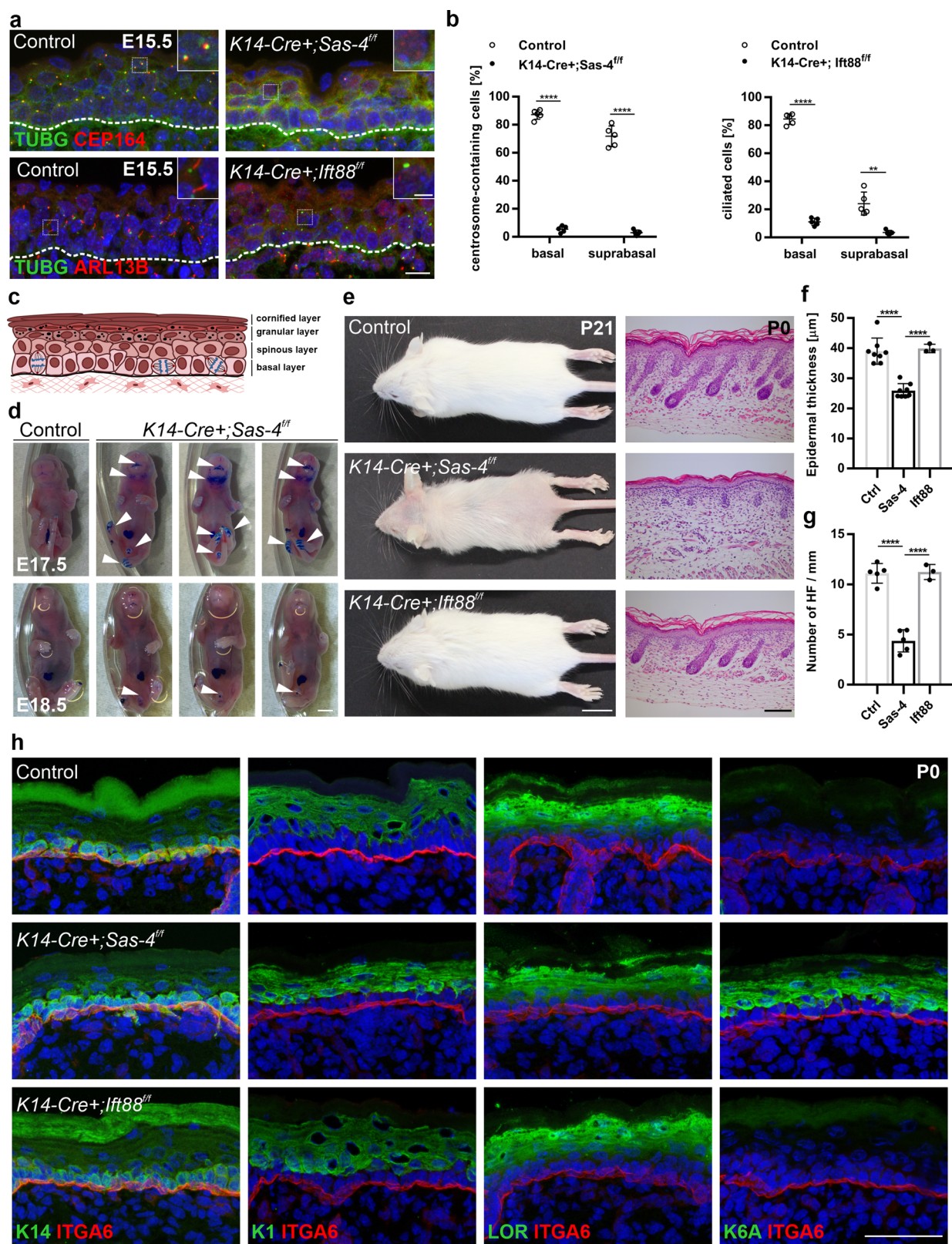

centrosome mutant epidermis (Fig. 1h). However, K6A, which is ectopically expressed in activated keratinocytes and in skin disease conditions, was markedly increased and indicated an abnormal response in the *Sas-4* mutant skin epithelium (Fig. 1h).

To test whether the secondary loss of cilia upon the removal of centrioles caused the centrosome-mutant phenotypes, we used

the *K14-Cre* line and a conditional allele of *Ift88*[31–33]. Immunostaining for TUBG and ARL13B (a marker for cilia) showed the presence of centrosomes but the loss of cilia in the *Ift88* mutant epidermis at E15.5 (Fig. 1a, b). Compared to *Sas-4*/centrosome mutants, we did not observe major skin phenotypes in the epidermis and hair follicles of *Ift88*/cilia mutants, as judged by morphological, histological and immunofluorescence

**Fig. 1 Centrioles, not cilia, are important for proper epidermal and hair follicle formation. a** Immunostaining of control and *K14-Cre+; Sas-4f/f* or *K14-Cre+; Ift88f/f* back-skin sections at E15.5 showing the loss of centrosome and centriole markers (TUGB, green) and (CEP164, red) or cilia marker (ARL13B, red). Insets are magnifications of the indicated dotted areas in each panel. The dashed line represents the epidermal-dermal interface in all panels (scale bar: 10 μm; inset: 3 μm). **b** Quantification of the centrosome- or cilia-containing cells in (**a**) with $n = 5$ independent animals. **p < 0.01, ****p < 0.0001 (two-tailed student's T-test). Bars represent mean ± SD (standard deviation). **c** Schematic of the epidermis depicting the different epidermal layers. **d** Toluidine Blue dye-penetration assay of Control and *K14-Cre+; Sas-4f/f* embryos at E17.5 and E18.5. Arrowheads indicate regions with delayed barrier formation in *K14-Cre+; Sas-4f/f* embryos (scale bar: 2 mm). **e** Gross phenotypes of Control, *K14-Cre+; Sas-4f/f* and *K14-Cre+; Ift88f/f* mice are shown at P21 (scale bar: 1 cm), as well as H&E histological staining of back-skin sections at P0 (scale bar: 100 μm). **f, g** Quantification of the interfollicular epidermal thickness (**f**), excluding the cornified layer, of Control ($n = 8$), *K14-Cre+; Sas-4f/f* ($n = 8$), *K14-Cre+; Ift88f/f* ($n = 3$) or the number of hair follicles (**g**) at P0 of Control ($n = 5$), *K14-Cre+; Sas-4f/f* ($n = 5$) and *K14-Cre+; Ift88f/f* ($n = 3$) mice. ****p < 0.0001 (two-tailed student's T-test or one-way ANOVA and Tukey's multiple comparisons test without adjustments). Bars represent mean ± SD. **h** Immunostaining of Control, *K14-Cre+; Sas-4f/f* and *K14-Cre+; Ift88f/f* back-skin sections at P0 stained for the epidermal layers' markers K14, K1 and Loricrin (LOR) (all green), the activation marker K6A and the basement membrane marker Integrin-α6 (ITGA6, red) (scale bar: 50 μm).

examination (Fig. 1e–h; Supplementary Fig. 1b, c), indicating that the centrosome mutant phenotypes are not due to the loss of cilia.

We concluded that, in contrast to cilia, centrioles and centrosomes are important to ensure normal development of the skin epithelium; however, despite the centriole deficiency in the basal keratinocyte progenitors, they showed robust regulation to allow the formation and maintenance of a generally functional skin barrier.

**The loss of centrioles in the epidermis leads to p53 upregulation and transient cell death.** Next, we used immunostaining to assess the molecular consequences of the loss of centrioles during epidermal development. The data showed that there was a gradual upregulation of nuclear p53 from E12.5-P0 in the *Sas-4*/centrosome mutant basal epidermis (Fig. 2a, b). In contrast, no upregulation of p53 was observed in the *Ift88*/cilia mutant epidermis at E15.5 (Supplementary Fig. 2a), indicating that cilia loss alone did not activate the same pathway[23,24]. In addition, cell death, marked by Cleaved-Caspase 3 (Cl.CASP3), was significantly increased in the *Sas-4* mutant epidermis at E15.5 (Fig. 2c, d). At P0, cell death was rarely detected while p53 was still high in the basal keratinocytes of centrosome mutants compared to controls (Fig. 2a, b, e). To check whether the high p53 in the centrosome mutant keratinocytes at P0 caused cell cycle arrest in G1 instead of cell death, similar to the reports in mammalian cell lines that lose centrioles in vitro[25,26,35], we performed cell cycle analyses on isolated epidermal keratinocytes. The cell cycle profiles showed no significant increase in the G1 population in centrosome mutant keratinocytes compared to controls (Fig. 2f).

Centriole loss causes prolonged mitotic duration which increases the mitotic index[23]. We quantified the number of mitotic cells, marked by phospho-Histone H3 (pHH3), in the basal layer of the epidermis of centrosome mutants and controls at E15.5 (Fig. 2g). We observed an increase in the mitotic index in the centrosome mutant epidermis compared to the control littermates (Fig. 2g)[23,24]. The delay in mitosis in centrosome mutants was also reflected in a slight, but not significant, increase in the G2/M keratinocyte population at P0 (Fig. 2f), likely because mitosis constitutes only a minor fraction of the total cell cycle (~3%).

To gain a deeper understanding of the transcriptional changes in epidermal keratinocytes during the early stages of centriole loss, we performed RNA-Seq analyses on micro-dissected and enzymatically separated back-skin epidermis of control and *Sas-4* mutants at E13.5, before any major phenotypes were apparent. Among the 258 differentially expressed genes, the vast majority (80%) were upregulated in centrosome mutant keratinocytes (Supplementary Table 1). As expected, gene ontology analyses showed that the most significantly over-represented category was

the "p53 downstream" signaling pathway (Fig. 2h). Consistent with the increased cell death in these mutants (Fig. 2c, d), the p53-downstream apoptosis target genes *Bbc3* (PUMA) and *Pmaip1* (NOXA) as well as the pro-apoptotic FAS ligand and BAX, were among the upregulated genes (Supplementary Table 1). In addition, p63 and p73 signaling pathways were over-represented among the upregulated genes, suggesting that the upregulated p53 in centrosome mutants might interfere with these closely related family members (Fig. 2h).

To study how the centrosome mutant keratinocytes coped with p53 upregulation without inducing cell death or major cell cycle changes at P0 (Fig. 2a–f), we performed RNA-Seq on isolated keratinocytes from centrosome mutants and controls and analyzed the changes in the transcriptional programs at birth. The data showed that over 3600 genes were differentially expressed at this stage, ~40% of which were upregulated in centrosome mutants (Supplementary Table 2). Besides the p53-downstream, p73, and p63 pathways, the EGFR (ErbB) and HIF1 pathways were also over-represented (Fig. 2i). In agreement with K6A upregulation (Fig. 1h), *Krt6a* and *Krt6b* were among the most highly induced genes in centrosome mutants, together with many epidermal differentiation-related genes (*S100a* and *Sprr* genes) (Supplementary Table 2). *Cdkn1a* (p21), which is induced upon the activation of the mitotic surveillance pathway in vitro, was also among the upregulated genes and was evident by immunostaining in the centrosome mutant epidermis at P0 (Supplementary Fig. 2b).

Both RNA-Seq experiments highlighted the perturbation in the p53/p63/p73 signaling axis in centrosome mutant keratinocytes, where more than 30% of the differentially expressed genes at E13.5 and ~20% at P0 have been shown to be targets of p63 in keratinocytes (Fig. 2j)[36]. Similar to controls, p63 and p73 were still readily detectable by immunostaining in the nuclei of basal cells in the centrosome mutant epidermis at P0 (Supplementary Fig. 2c), suggesting that the unusually high levels of p53 in the nuclei of centrosome mutant keratinocytes might compete with p63 and p73 for target binding.

**Centrosome mutants activate the mitotic surveillance pathway.** Centrioles are required for efficient mitosis, which is associated with the suppression of the activation of the p53-dependent mitotic surveillance pathway (Fig. 3a)[30]. We tested whether the loss of p53 can bypass the activation of this pathway in the centrosome mutant skin epidermis (Fig. 3b). We observed that the simultaneous knockout of *p53* in the *Sas-4*/centrosome mutant skin epidermis significantly rescued the gross epidermal and hair follicle defects, including the epidermal thickness and hair follicle numbers (Fig. 3b–d). Moreover, the increase in K6A expression in centrosome mutant keratinocytes was dependent on p53, because the *Sas-4 p53* double mutant epidermis showed a

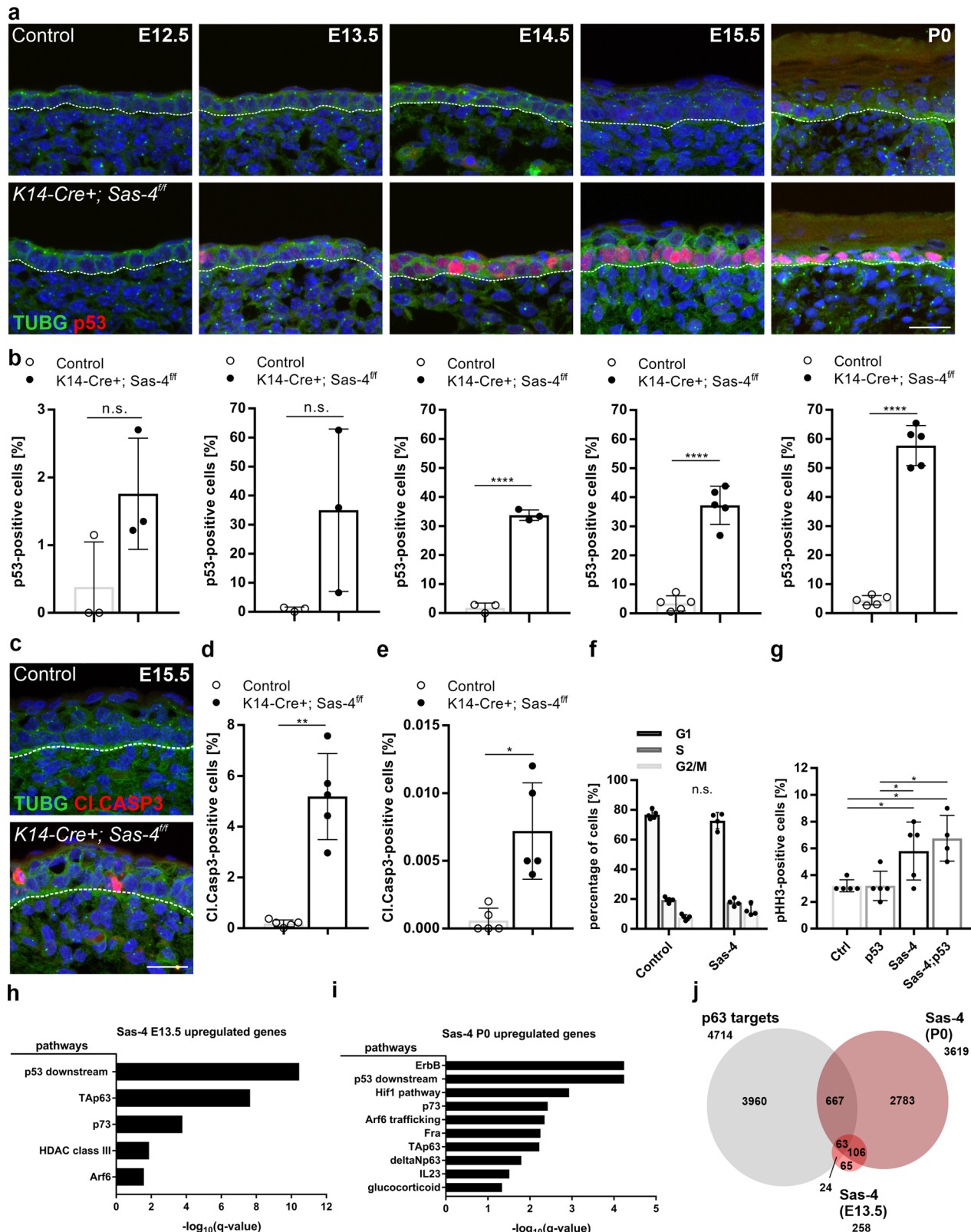

rescue of K6A back to the levels in the control (Fig. 3e). However, the loss of p53 does not rescue cilia formation or prolonged mitosis, because it does not restore centrioles or shorten mitosis[23], and thus, the double mutants still showed a higher mitotic index compared to controls (Fig. 2g).

We then asked whether the loss of *Usp28* or *53bp1*, which have been implicated in the activation of the mitotic surveillance pathway in human cell lines in vitro (Fig. 3a)[27–29], , would recapitulate the p53 deficiency in centrosome mutants in vivo. We generated *Usp28* and *53bp1* mutant mice using CRISPR/Cas9 (see "Methods") and crossed each allele to the *Sas-4* conditional mutants and *K14-Cre* to generate *Sas-4 Usp28* and *Sas-4 53bp1* double mutant skin epidermis. Remarkably, our data showed that these mice had a phenotypic rescue

**Fig. 2 Upregulation of p53 in the centrosome mutant epidermis. a** Immunostaining of Control and *K14-Cre+; Sas-4^{f/f}* back-skin sections at the indicated stages showing the loss of the centrosome marker (TUBG, green) and gradually increase in nuclear p53 (red) in the mutant mice (scale bar: 20 μm). The dashed line represents the epidermal-dermal interface in all panels. **b** Quantification of the percentage of p53-positive nuclei in the back-skin basal epidermis of control and *K14-Cre+; Sas-4^{f/f}* mice at the corresponding stage shown in (**a**). $n = 3$ for controls and *K14-Cre+; Sas-4^{f/f}* from E12.5–E14.5 and $n = 5$ for Controls and *K14-Cre+; Sas-4^{f/f}* at E15.5 and P0. Bars represent mean ± SD in these and subsequent graphs. ns not significant, ****$p < 0.0001$ (two-tailed student's *T*-test). **c, d** Immunostaining (**c**) and quantification (**d**) of Control ($n = 5$) and *K14-Cre+; Sas-4^{f/f}* ($n = 5$) back-skin sections at E15.5 for cell death (Cl.CASP3, red) in the mutant epidermis (scale bar: 20 μm). **e** Similar to (**d**) but at P0 with $n = 5$ for each genotype. **d, e** *$p < 0.05$, **$p < 0.01$ (two-tailed student's *T*-test). **f** Cell cycle profiles of isolated primary keratinocytes at P0 of Control ($n = 3$) and *K14-Cre+; Sas-4^{f/f}* ($n = 3$) mice. ns not significant (two-tailed student's *T*-test). **g** Quantification of the percentage of pHH3-positive cells in the basal layer of back-skin epidermal sections at E15.5 of Control ($n = 5$), *K14-Cre+; Sas-4^{f/f}* ($n = 5$), *K14-Cre+; Sas-4^{f/w}; p53^{f/f}* ($n = 5$) and *K14-Cre+; Sas-4^{f/f}; p53^{f/f}* ($n = 4$) mice. ns not significant, *$p < 0.05$ (two-tailed student's *T*-test). **h** Bar chart depicting the over-represented pathways (from the pathway interaction database (PID)) of differentially expressed genes in the *K14-Cre+; Sas-4^{f/f}* epidermis compared to controls at E13.5. The *X*-axis represents the negative $\log_{10}$ of the *q*-values, the adjusted p-values optimized by the false discovery rate. **i** Similar to (**h**) but for P0 epidermal keratinocytes. **j** Venn diagram showing the overlap of published p63 target genes[36] with differentially expressed genes of *K14-Cre+; Sas-4^{f/f}* at E13.5 and P0.

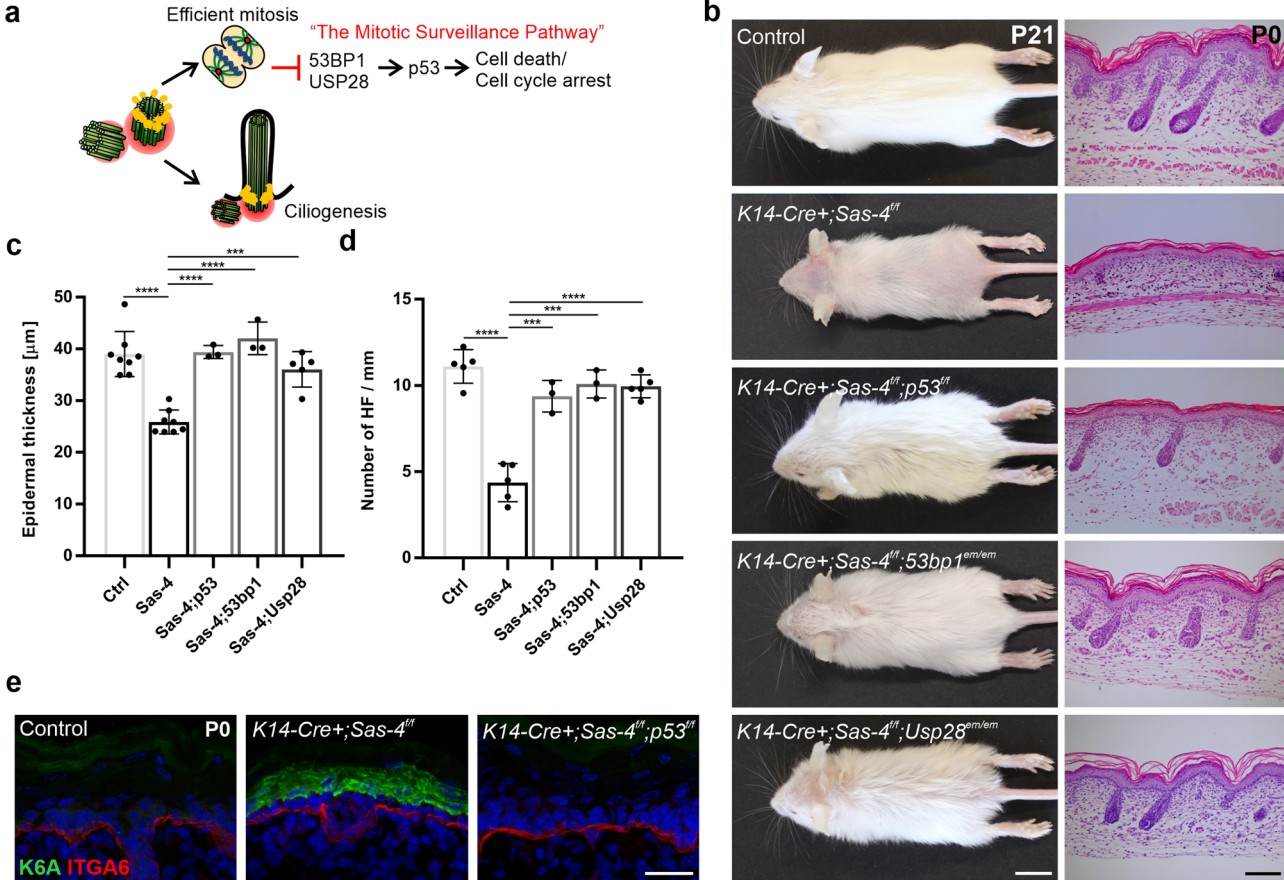

**Fig. 3 The skin epidermal phenotypes in centrosome mutants are due to the activation of the mitotic surveillance pathway. a** Centrioles, besides providing the essential template for cilia, ensure efficient mitosis and repress the activation of the mitotic surveillance pathway: p53 stabilization mediated by 53BP1 and USP28 leading to cell death (mainly shown in vivo) or cell cycle arrest (mainly shown in vitro). **b** Gross phenotypes of Control, *K14-Cre+; Sas-4^{f/f}*, *K14-Cre+; Sas-4^{f/f}; p53^{f/f}*; *K14-Cre+; Sas-4^{f/f}; 53bp1^{em/em}* and *K14-Cre+; Sas-4^{f/f}; Usp28^{em/em}* mice are shown at P21 (scale bar: 1 cm), as well as H&E histological staining of back-skin sections at P0 (scale bar: 100 μm). **c** Quantification of the interfollicular epidermal thickness (**c**), excluding the cornified layer, at P0 of Control ($n = 8$), *K14-Cre+; Sas-4^{f/f}* ($n = 8$), *K14-Cre+; Sas-4^{f/f}; p53^{f/f}* ($n = 3$), *K14-Cre+; Sas-4^{f/f}; 53bp1^{em/em}* ($n = 3$) and *K14-Cre+; Sas-4^{f/f}; Usp28^{em/em}* ($n = 5$) mice. Control and *K14-Cre+; Sas-4^{f/f}* animals are part of the same experiment in Fig. 1. **d** Quantification of the number of hair follicles at P0 of Control ($n = 5$), *K14-Cre+; Sas-4^{f/f}* ($n = 5$), *K14-Cre+; Sas-4^{f/f}; p53^{f/f}* ($n = 3$), *K14-Cre+; Sas-4^{f/f}; 53bp1^{em/em}* ($n = 3$) and *K14-Cre+; Sas-4^{f/f}; Usp28^{em/em}* ($n = 5$) mice. **c, d** ***$p < 0.001$, ****$p < 0.0001$ (two-tailed student's *T*-test or one-way ANOVA and Tukey's multiple comparisons test without adjustments). Bars represent mean ± SD. **e** Immunostaining of Control, *K14-Cre+; Sas-4^{f/f}* and *K14-Cre+; Sas-4^{f/f}; p53^{f/f}* back-skin sections at P0 for K6A (green) and ITGA6 (red) (scale bar: 25 μm).

similar to the *Sas-4 p53* double mutants, establishing their essential role in the p53-dependent pathway in vivo (Fig. 3b–d). These rescues were quite significant even though our CRISPR/Cas9-generated mutations in *Usp28* and *53bp1* had residual proteins by immunostaining (Supplementary Fig. 3a, b),

suggesting that they were likely to be hypomorphic alleles. Consistent with the variable nature of hypomorphic mutations, p53 nuclear levels only significantly decreased in the *Sas-4 53bp1* double mutants compared to the *Sas-4* single mutants (Supplementary Fig. 3d).

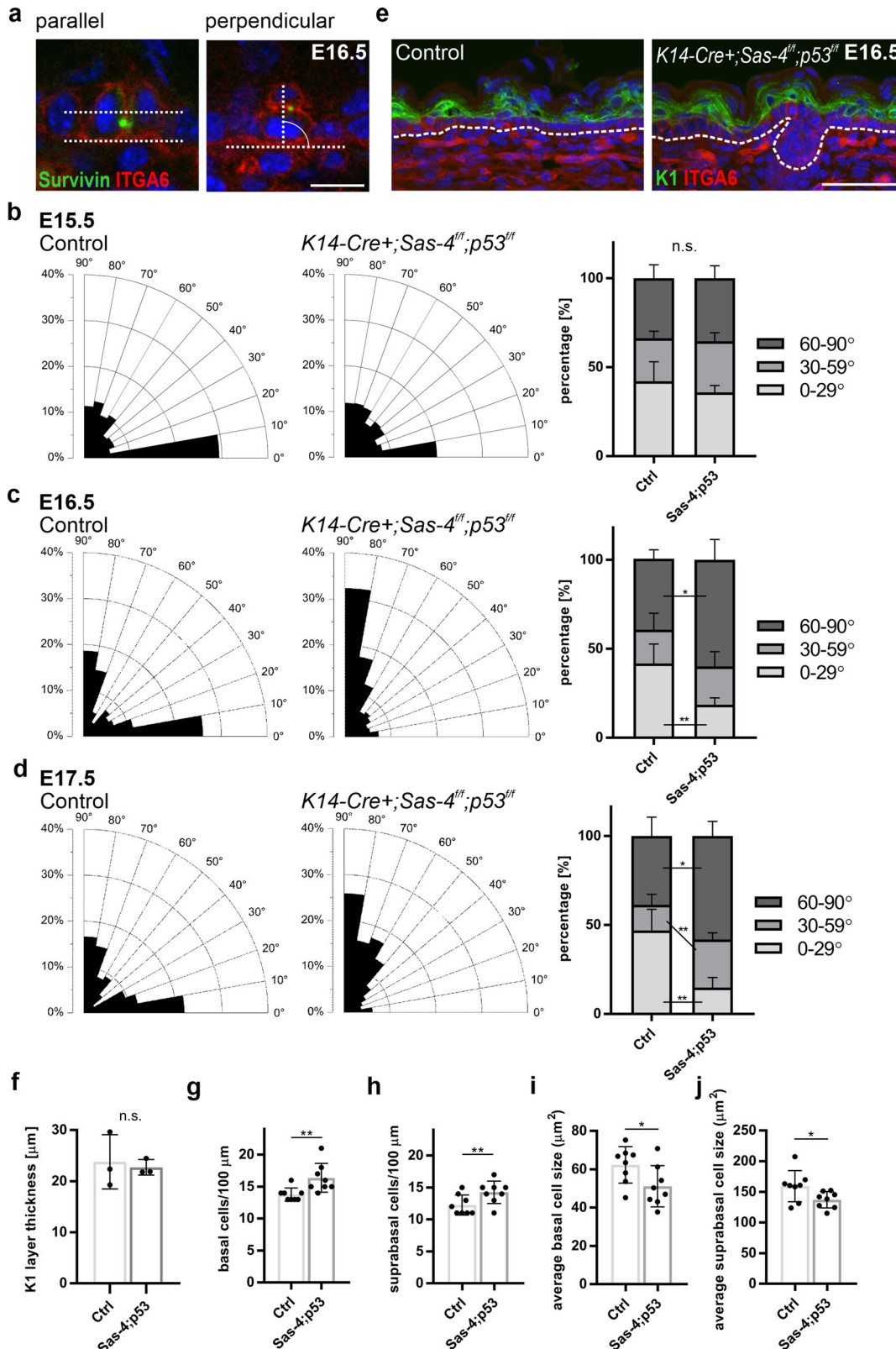

**Cell division orientation of the basal progenitors does not correlate with epidermal differentiation in the mutants**. The *Sas-4*/centrosome *p53* double mutants, which showed normal epidermal thickness at birth (Fig. 3b, c), allowed us to examine the consequences of the loss of centrioles in regulating cell division orientation and its relationship to epidermal differentiation. We used immunostaining for Integrin-α6 (ITGA6), to mark the basement membrane zone, and Survivin, a mid-body marker to highlight the mitotic cells in late anaphase to telophase when the division orientation is final and measured the angles of cell division orientation in sections of the back-skin of *Sas-4 p53* double mutant and control animals (Fig. 4a–d)[12]. Published data that analyzed late stages of mitosis considered E16–E17 as the peak of epidermal stratification, with a fine balance between

**Fig. 4 The proportions of cell division orientation in the basal progenitors do not correlate with differentiation at late stages in *Sas-4 p53* mutants.**
**a** Representative image of parallel and perpendicular dividing basal cells from immunostaining (Survivin, green, and ITGA6, red) of back-skin sections of the interfollicular epidermis of Controls at E16.5 (scale bar: 10 µm). **b** Radial histograms of the distribution of the angles of cell division orientation in late anaphase to telophase of basal epidermal cells of Controls and *K14-Cre+; Sas-4^f/f^; p53^f/f^* animals at E15.5. Percentages of parallel (0–29°), oblique (30–59°), and perpendicular (60–90°) dividing basal layer cells of Controls (*n* = 3) and *K14-Cre+; Sas-4^f/f^; p53^f/f^* (*n* = 3) animals at E15.5. **c** Similar to (**b**) at E16.5. **d** Similar to (**b**) at E17.5 with *n* = 5 independent animals for each genotype and time point. **b–d** ns not significant, *p < 0.05, **p < 0.01 (two-tailed student's *T*-test). **e**, **f** Immunostaining (**e**) and quantification (**f**) of the differentiated (K1-positive, green) layer in back-skin sections at E16.5 of Control (*n* = 3) and *K14-Cre+; Sas-4^f/f^; p53^f/f^* (*n* = 3) mice (scale bar: 50 µm). The dashed line represents the epidermal-dermal interface in all panels. **g**, **h** Quantification of the density (cells per 100 µm) of basal (**g**) and suprabasal (**h**) layer cells of sagittal back-skin epidermal sections of Controls (*n* = 8) and *K14-Cre+; Sas-4^f/f^; p53^f/f^* (*n* = 8) animals at E15.5. **i**, **j** Quantification of the average cell size of basal (**i**) and suprabasal (**j**) layer cells from (**g**) and (**h**), respectively, with *n* = 8 independent animals for each genotype. ns not significant, *p < 0.05, **p < 0.01 (two-tailed student's *T*-test). Angle measurements were compared using a two-way ANOVA, Kolmogorov–Smirnov test, and Chi-squared test, all of which gave similar statistical significance outcomes. Bars represent mean ± SD.

parallel (0–29°) and perpendicular (60–90°) division orientation, whereas earlier stages (E14–E15) showed random orientation proportions[18]. Consistent with this, at E15.5, there was no significant difference between the cell division orientation proportions between control and *Sas-4 p53* mutants (Fig. 4b). Statistical analyses (two-way ANOVA and Kolmogorov–Smirnov test) showed that the three division orientation bins (parallel, oblique, or perpendicular) had an equal likelihood of occurrence regardless of the genotype, suggesting that they are random in nature.

However, at E16.5 and E17.5, the measurements showed a significant shift towards more perpendicular cell division orientation in the *Sas-4 p53* double mutant skin epidermis (~60%) at both E16.5 and E17.5 compared to heterozygous and *p53* mutant controls (~40%) (Fig. 4c, d; Supplementary Fig. 4a). The increase in the proportion of perpendicular division orientation in the mutants was accompanied by a significant decrease in parallel division orientation at both stages (~15–20% in the mutants compared to ~40% or more in controls) (Fig. 4c, d). In contrast, similar measurements in *Sas-4* single mutants at E16.5, which have a thinner epidermis by P0 (Fig. 1f), had an increased proportion of parallel divisions (Supplementary Fig. 4b), in line with the prevailing model (see "Discussion").

We next tested if the cell division orientation data in *Sas-4 p53* double mutants at E16–E17 were consistent with the current model of basal keratinocyte progenitors' division orientation determining the balance between proliferation and differentiation. First, if a perpendicular division orientation was asymmetric in nature, then an increase in its proportion should lead to more differentiation in the mutants. Thus, we measured the thickness of the K1-positive suprabasal layers at E16.5, and the data showed that it was not changed in the mutants relative to controls (Fig. 4e, f). In addition, the skin epidermal histology and thickness in the *Sas-4 p53* double mutant skin at birth were also similar to controls (Fig. 3b, c). Second, if a parallel division orientation was symmetric, then a decrease in its proportion should result in a lower density of basal keratinocyte progenitors. Instead, the *Sas-4 p53* double mutant skin epidermis showed an increase in the density of basal, as well as suprabasal, cells compared to controls (Fig. 4g, h). Accordingly, the average cell size in each layer decreased proportionally in the double mutants (Fig. 4i, j). Both phenotypes of cell number and average size changes in the *Sas4 p53* double mutants seemed to be dependent on p53 and/or cilia (Supplementary Fig. 4c, d). We then checked whether increased proliferation could explain the maintenance, or even increase, in the number of basal cells in the mutants, but the data using EdU incorporation showed no significant difference between the mutants and controls (Supplementary Fig. 4e). In this context, our data suggested that basal keratinocyte progenitors' proliferation and differentiation were uncoupled from cell division orientation during the later stages of epidermal skin development.

**Cell delamination may contribute to epidermal stratification.** Because our data suggested that basal progenitor division orientation may not necessarily dictate epidermal stratification and differentiation, we next asked whether other mechanisms, such as cellular delamination, contribute to stratification during development. Embryonic development on the organismal level is reliant on proliferation and cell division in vivo. Therefore, we used a well-established ex vivo developing mouse skin culture that recapitulates epidermal differentiation as well as hair follicle development[37]. We treated skin explant cultures at E13.5, which had only a thin K10-positive differentiated layer, with mitomycin C (MMC) to stop proliferation and cell division, and then incubated the explants for one (E14.5) or two (E15.5) days[38]. Proliferation and cell division were drastically inhibited in the MMC-treated skin explants compared to controls, as assayed by EdU incorporation and pHH3 staining at E14.5 and E15.5 (Supplementary Fig. 5a). We observed some speckle nuclear staining of pHH3 in MMC-treated skin, presumably in cells arrested in G2, but no staining reflecting condensed mitotic chromosomes. Importantly, despite the inhibition cell division and elevated cell death (Cl.CASP3, Supplementary Fig. 5b), the MMC-treated skin epidermis retained some capacity to stratify and differentiate, as shown by the substantial increase in the K10-positive layer thickness at E14.5 and E15.5 (Supplementary Fig. 5a, c). Moreover, the number of basal (K10-negative) cells sharply decreased in the MMC-treated explants compared to the corresponding controls at E14.5 and E15.5 (Supplementary Fig. 5d). In addition, the number of K10-expressing suprabasal cells in the MMC-treated explants increased approximately fourfold at E14.5 and E15.5 compared to the E13.5 controls but was still significantly lower than that in the vehicle-treated controls (Supplementary Fig. 5e). The combined cell number deficiency in the MMC-treated skin cultures was not unexpected because of the inhibition of proliferation and resulting in cell death (Supplementary Fig. 5a, b). The data suggested that upon the inhibition of proliferation and cell division, other mechanisms, most likely cellular delamination from the basal to the suprabasal layers, can contribute to epidermal stratification and differentiation[18].

**Time-lapse imaging reveals epidermal basal layer flexibility, cellular delamination, and suprabasal divisions.** We then asked whether a perpendicular cell division orientation and/or cellular delamination of the basal progenitors contribute to epidermal stratification and the formation of the suprabasal layers. We recorded time-lapse movies of the skin explants on filters between E13.5–E15.5;[39] however, the *Z*-dimension resolution was not sufficient to follow dividing cells (Supplementary Fig. 5f). Thus, we flipped the explants as skin rolls to transform the initial *Z*-dimension into an *XY* plane (see "Methods"). This culture method did not affect skin epidermal stratification and

differentiation as shown by K10 staining and even better resembled the embryonic skin in vivo than the flat skin cultures (compare the staining on skin rolls in Supplementary Fig. 5g and the flat skin explants in Supplementary Fig. 5h). We focused on the daughter cells of perpendicularly dividing basal progenitors (~80–90°) that point away from the epidermal-dermal interface, and assessed their position over time. Our data showed that almost half of these cells could be incorporated close to the neighboring basal cells (Fig. 5a; Supplementary Movie 1), whereas the other half remained in the second layer (Fig. 5b; Supplementary Movie 2). Moreover, we observed progenitor basal cells that delaminated, moved up, and divided suprabasally shortly afterward (Fig. 5c; Supplementary Movie 3). The data from the time-lapse experiments suggested that the progenitors in the basal layer can stratify by delamination and may be boosted by suprabasal divisions to form the multi-layered epidermis, most likely independent of cell division orientation.

During our analyses of the developing mouse embryonic skin epidermis, we noticed that the basal layer character does not necessarily require that the basal keratinocytes appear in direct contact with the underlying basement membrane. This was particularly evident upon staining with ITGA6 (basal) and K10 (suprabasal) at different epidermal developmental stages (Fig. 5d). From E13.5–E17.5, ITGA6 was surrounding the entire basal keratinocytes, including some that seemed to be in the second layer but were negative for K10 (Fig. 5d), as well as the daughter cells in a perpendicular division that point away from the epidermal-dermal interface (Fig. 4a)[40]. The ex vivo flat skin explant culture on filters at E13.5, which was just released from the stretching tension of the embryo, showed an even thicker (approximately three cell layers) ITGA6-positive and K10-negative pseudo-stratified basal layer at E13.5 (Supplementary Fig. 5h), supporting the in vivo data. On the other hand, basement membrane markers, such as COL4A4 and LAMA1, were confined to the basement membrane at E16.5 (Supplementary Fig. 5i, j). Collectively, our data suggested that the skin epidermal basal layer is more flexible during development where cells move to a seemingly second layer, for example during a perpendicular division, but still retain a basal keratinocyte progenitor character.

**The developing epidermis stratifies and differentiates in two phases**. To gain more insight into the epidermal stratification process at the different stages, we turned to a modeling-based approach ("Supplementary theory" in "Methods") informed by measurements of dynamic tissue growth parameters during back-skin interfollicular epidermal development (E12.5–E18.5), in the anteroposterior axis (sagittal sections, Fig. 6) and the dorso-ventral axis (transverse sections, Supplementary Fig. 6). We quantified the following parameters: (1) the net rate of embryonic growth (Fig. 6a; Supplementary Fig. 6a, b), (2) the density of basal and suprabasal cells (Fig. 6b, c; Supplementary Fig. 6c, d), (3) the proliferation index of basal and suprabasal cells (Fig. 6d; Supplementary Fig. 6e). Based on our data above (Fig. 5d), we defined basal progenitor cells by their expression of ITGA6 but not K10, whereas suprabasal cells express K10 but not ITGA6. While the total embryonic growth was approximately linear along the different axes between E12.5–E18.5 (Fig. 6a; Supplementary Fig. 6a, b), we made two important observations regarding the suprabasal layer. First, the major increase in the number of suprabasal cells occurred between E13.5–E15.5 (Fig. 6b, c; Supplementary Fig. 6c, d). Second, the suprabasal cells were highly proliferative during the same window of epidermal development (Fig. 6d; Supplementary Fig. 6e), consistent with the data from the time-lapse imaging of skin roll explants (Fig. 5c; Supplementary

Movie 3)[10,41,42]. Overall, our measurements and modeling support a two-phase behavior for basal and suprabasal cells during epidermal stratification and differentiation (Fig. 6e). The first phase between E12.5–E15.5 is characterized by proliferation and amplification to fuel stratification, where the suprabasal cells proliferate to populate the newly-forming suprabasal layers. Consistently, the total number of cells rises exponentially by a factor of 7.4 during this phase (Fig. 6e), almost twice the net area expansion of the tissue (Fig. 6a; Supplementary Fig. 6a, b). In addition, the density of suprabasal cells reaches parity and equilibrium with that of the basal progenitor cells at the end of this phase (at E15.5; Fig. 6c; Supplementary Fig. 6d). The second phase between E15.5–E18.5 is characterized by a precipitous slowdown in the proliferation rate, especially in the suprabasal cells which largely undergo terminal differentiation (Fig. 6d; Supplementary Fig. 6e). The total number of cells in this second phase rises linearly by a mere factor of 2 (Fig. 6e), in proportion to the net expansion of the tissue. The cell densities in both layers remain similar and largely unchanged during this phase (Fig. 6c; Supplementary Fig. 6d). Collectively, our data suggest that the major phase of epidermal stratification takes place between E13.5–E15.5 and is boosted by the proliferating suprabasal cells[10,41,42].

**The human skin epidermis harbors a proliferative suprabasal population that is increased in hyperproliferative disorders**. Our data on the high proliferation rate of suprabasally-committed keratinocytes prompted us to further investigate whether the adult human skin also harbors such a population and assess if it is more prominent in hyperproliferative disorders of the skin[43–45]. Immunostaining using the proliferation marker Ki67 together with K10 showed that ~3% of the K10-positive suprabasal cells in the normal human skin were proliferative (Fig. 7a). As expected in hyperproliferative skin diseases, such as psoriasis and atopic dermatitis, the percentage of proliferative cells was increased approximately fourfold in the basal layer, and so did that of the suprabasally-proliferative population, which increased approximately twofold (Fig. 7a). The data indicated that the human epidermis has a proliferative suprabasal population that is increased, along with the hyperproliferative basal population, in certain disease conditions.

## Discussion

How simple epithelia transform into complex stratified barriers, and whether the stratification and differentiation programs are dependent on a shift to a perpendicular division orientation of the progenitors, are still open questions in epithelial biology. Here, we use centrosome mutants to disrupt cell division orientation and test its relationship with skin epidermal differentiation.

In the developing *Sas-4* mutant epidermis, the basal keratinocyte progenitors that lose centrioles upregulate p53 and only a small fraction of the cells die (~5% with Cl.CASP3) (Fig. 2a–d). In newborns, the basal acentriolar keratinocytes also show high levels of p53, but only rare cells die (Fig. 2e). In contrast, centriole loss in the early embryo and developing brain leads to widespread cell death and tissue degeneration[23,24], whereas centriole loss in mammalian somatic cell lines leads to an irreversible G1 cell cycle arrest[25,26,35]. The data indicate that the skin epidermal keratinocytes are more robust than early embryonic and developing brain cells, or even certain cell types in culture, suggesting that basal keratinocytes adapt to centriole loss and the high levels of p53 by regulating their transcriptional programs to maintain a relatively intact skin barrier (Figs. 1d and 2). Our findings are consistent with the skin phenotypes not being prominent features in humans with mutations in genes encoding centrosomal

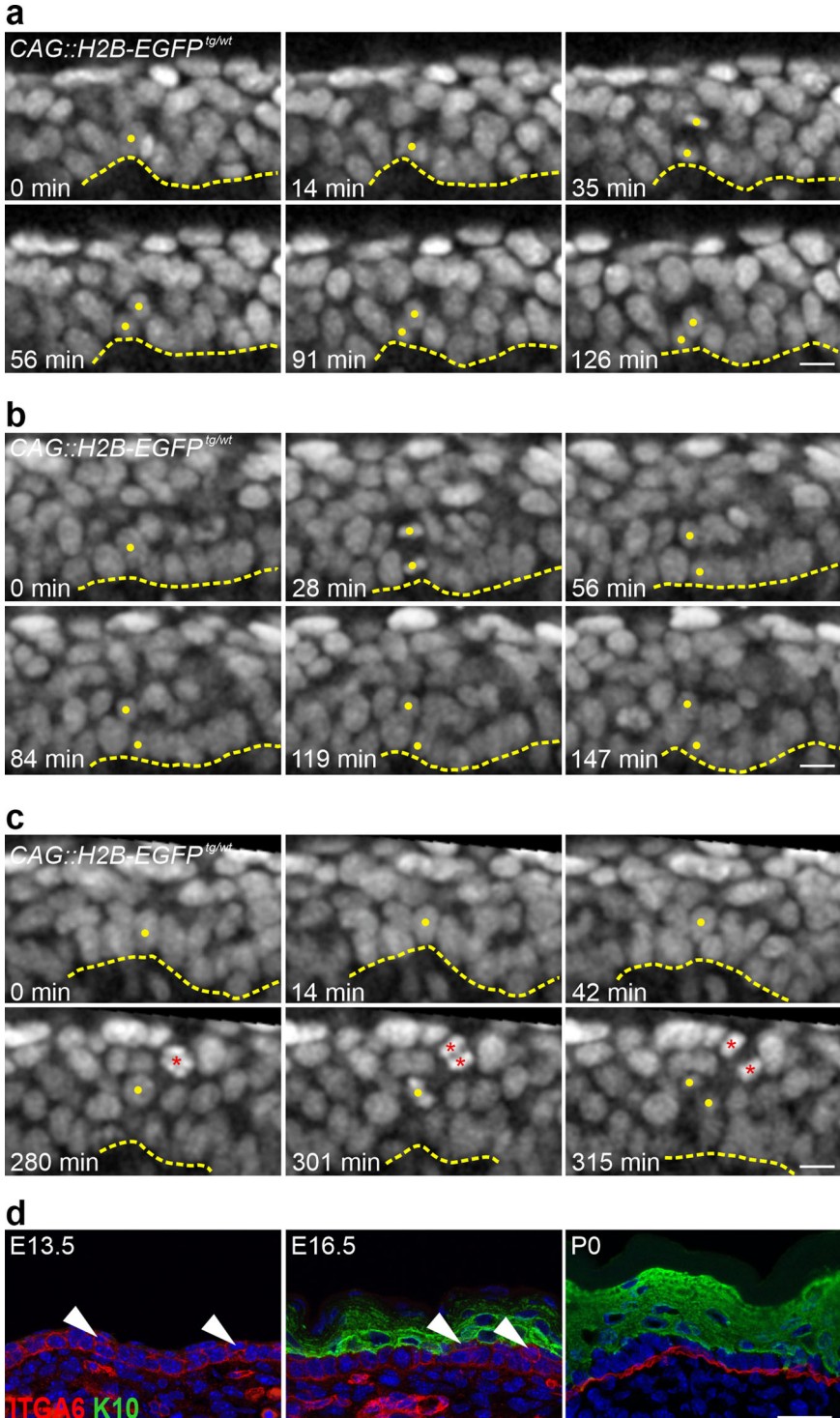

**Fig. 5 The skin epidermal basal layer is more flexible during development. a–c** Time-lapse images of the CAG::H2B-EGFP nuclear reporter (gray) in E13.5–E14.5 skin-roll explants taken every 7 min. **a, b** Examples of perpendicular dividing basal cells with daughter cells that point away from the epidermal-dermal interface that was either integrated into the basal layer after division ((**a**), 44 out of 94 cells), or remained in the second layer for the duration of the analysis ((**b**), 50 out of 94 cells). **c** An Example of a delaminating cell that left the basal layer moved suprabasally and divided (a total of 46 cells observed). Note the other dividing suprabasal cell (red asterisks) (scale bars: 10 μm). The data were obtained from five explants taken from five different embryos in five independent imaging experiments. The dashed line represents the epidermal-dermal interface in all panels. Yellow dots mark the cells of interest. **d** Immunostaining of back-skin sections at the indicated stages stained for ITGA6 (red) and K10 (green) (scale bar: 25 μm). Arrowheads indicate cells in the second layer that are positive for ITGA6 but negative for K10.

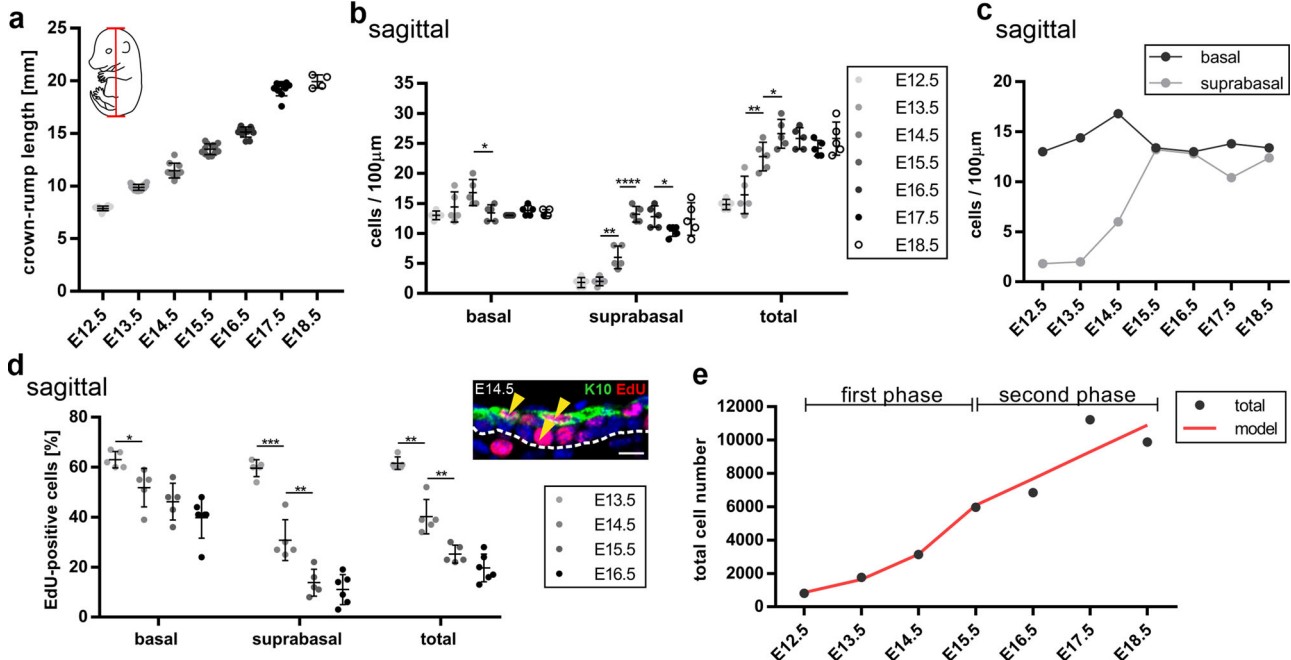

**Fig. 6 Skin epidermal stratification occurs in two phases. a** Quantification of the crown-rump distance as depicted in the embryo schematic (red line) in wild-type embryos from E12.5–E18.5. Note the linear trend of the growth. E12.5: $n = 10$, E13.5: $n = 16$, E14.5: $n = 10$, E15.5: $n = 12$, E16.5: $n = 11$, E17.5: $n = 10$ and E18.5: $n = 4$ **b, c** Cell densities (cells per 100 μm) of basal and suprabasal layers of sagittal back-skin sections of the epidermis from E12.5–E18.5 with $n = 5$ independent animals for each time point. **c** The suprabasal cell density reaches parity with that of the basal layer starting at E15.5. **d** Quantification of EdU-positive cells in the basal and suprabasal layers from sagittal back-skin sections of the epidermis at the indicated time points, with an example of the EdU (red)/K10 (green) staining shown at E14.5 (arrowheads) (scale bar: 10 μm). $n = 5$ for E13.5–E15.5 and $n = 6$ for E16.5. **e** The graph depicts the exponential increase in total cell number in the first phase of stratification between E12.5–E15.5 and the linear-trend increase in the second phase between E15.5–E18.5, shown as the calculated total cell number from the measurements (total) and in agreement with the model (red line). $*p < 0.05$, $**p < 0.01$, $***p < 0.001$, $****p < 0.0001$ (two-tailed student's $T$-test). Bars represent mean ± SD.

proteins[22]. The Café au Lait skin pigmentation defect is a rarely reported skin phenotype in these individuals[46,47]. These skin pigmentation defects are likely due to high p53 and the tanning effect[48].

The skin epithelial centrosome mutants have a thin epidermis and sparse hair (Fig. 1e–f). These two prominent centrosome-associated phenotypes are dependent on the activation of the mitotic surveillance pathway because they are rescued upon the removal of p53, 53BP1, or USP28 (Fig. 3a–d). Given that p63 is the master transcription factor governing epidermal stratification and skin appendage development[4], and that p53 and p63 share a consensus DNA binding site, our RNA-Seq data (Fig. 2h–j; Supplementary Tables 1 and 2) suggest that the abnormal increase in p53 levels in the developing skin epithelium of centrosome mutants disrupts p63 functions leading to the skin defects. In addition, the rescue of the activation of the mitotic surveillance pathway using *53bp1* or *Usp28* mutations in the centrosome mutant skin epithelium confirms the conservation of this pathway in mammalian tissues in vivo[49,50].

Many signaling pathways have been associated with centrosomes and their extensions, the primary cilia[51]. The phenotypes of the skin epithelial centrosome double mutants (Fig. 3b–d), which also lack cilia, and *Ift88*/cilia mutants (Fig. 1e–g), confirm earlier reports about the requirement of cilia and cilia-associated signaling mainly in postnatal hair follicles during homeostasis[32]. Our genetic data do not support a major link between centrosomes or cilia and Notch signaling in the skin epithelium[33], where Notch signaling plays roles in epidermal and hair shaft differentiation[6,41,52].

During the first phase of epidermal development between E13–E15, our data show that the daughter cells of the basal

progenitors that point away from the epidermal-dermal interface in a perpendicular division can be incorporated in the basal layer following division (Fig. 5a, Supplementary Movie 1). Moreover, the inhibition of cell division in the developing skin does not completely abolish cellular differentiation and stratification and may be compensated by other mechanisms (Supplementary Fig. 5a–e). Indeed, our time-lapse imaging data show that cellular delamination from the basal layer is a possible mechanism of stratification (Fig. 5c; Supplementary Movie 3), with published evidence supporting or excluding this mechanism in the epidermis[18,53–56]. The delaminating cells may boost stratification because they divide right after delamination to the suprabasal layers. In addition, cortical tension and cell adhesion have been shown to regulate the balance between cell proliferation and differentiation in skin epidermal keratinocytes in vitro[55]. The fast-expanding skin in the developing embryo is under tension and the tissue tension appears to be released in the skin explant cultures. However, the epidermis in the explants retains the ability to stratify and differentiate (Supplementary Fig. 5g, h), particularly in the skin roll cultures that are embedded in a matrix (see "Methods"). The skin roll explants seem to better recapitulate the epidermal stratification program in vivo (Supplementary Fig. 5g), perhaps due to the pressure offered by the embedding matrix. The difference between the two skin explant methods suggests that stratification and differentiation may also be uncoupled. Together with the time-lapse imaging data, our findings suggest that local cell-cell interactions and tension differentials, rather than global tissue tension, might dictate cell fate choices within the basal layer of the epidermis.

During the second phase of epidermal development between E15.5–E17.5, the *Sas-4 p53* mutants show a decrease in the

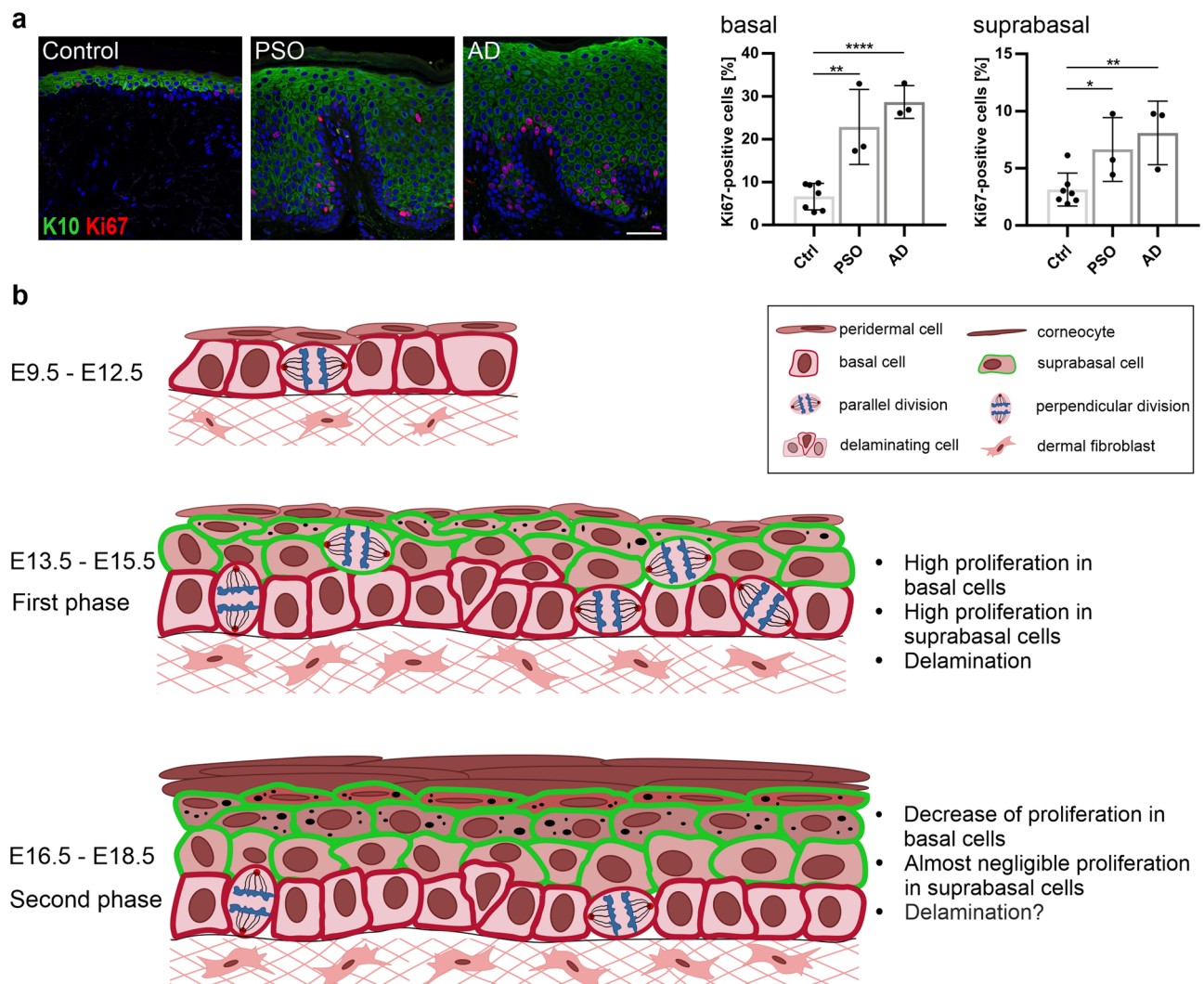

**Fig. 7 The human epidermis harbors a proliferative suprabasal population and a proposed model of mouse skin epidermal development. a** Immunostaining of Control ($n = 7$), psoriasis (PSO) ($n = 3$) and atopic dermatitis (AD) ($n = 3$) human skin samples for Ki67 (red) and K10 (green) (scale bar: 50 μm), and quantification of the percentage of Ki67-positive cells in the basal and suprabasal layers. *$p < 0.05$, **$p < 0.01$, ****$p < 0.0001$ (two-tailed student's T-test). Bars represent mean ± SD. **b** An illustration of the model for mouse skin epidermal development. The simple-like epithelium between E9.5–E12.5 undergoes its main stratification events during the first phase between E13.5–E15.5, where both basal and suprabasal keratinocytes have high proliferation/division rates and basal cells delaminate to fuel stratification. In the second phase between E15.5–E18.5, proliferation rates are slower, decreasing considerably in the suprabasal layer, delamination may be a dominant mechanism in maintaining stratification during tissue growth, and the angle of cell division orientation in the basal keratinocytes may be uncoupled from epidermal differentiation.

fraction of parallel division orientation and an increase in the perpendicular one in basal keratinocyte progenitors that do not lead to a corresponding decrease in basal layer density, or a suprabasal layer thickening (Fig. 4e–g), as predicted by the model that cell division orientation and epidermal differentiation are coupled[57]. Of note, the increase in cell densities and cell packing defects in both layers of the epidermis in *Sas-4 p53* double mutants (Fig. 4g–j) seems to be dependent on *p53* and/or cilia (Supplementary Fig. 4c, d), and importantly, the *p53* single mutants do not significantly shift their cell division orientation relative to controls at E16.5 (Supplementary Fig. 4a). It is interesting to note that the proportional increase in cell densities in both layers in these mutants at E16.5 is consistent with our data suggesting an equilibrium and a counting mechanism across the epidermal layers during this stage (Fig. 6c; Supplementary Fig. 6d). Collectively, our data suggest that a perpendicular division of the basal progenitors during this phase does not necessarily result in an asymmetric cell fate. It has been known

for a few decades that switching primary mouse skin epidermal keratinocytes in culture to high calcium levels induces differentiation independent of cell division[58]. The data suggest that commitment to differentiation in the basal progenitors is a multi-step process that is independent of division orientation and is associated with cellular detachment or delamination, similar to earlier reports in the developing and adult mouse skin epidermis as well as in human skin equivalents[54–56]. It is worth noting that perpendicular divisions have only been rarely observed in the adult skin epidermis[54,59].

It has recently been shown that the cellular geometry of the epidermal basal progenitors correlates with cell division orientation in different body sites in the developing embryo at E14.5[60]. However, all the skin body sites generate a stratified epidermis and a fully functional barrier by E17.5 (Fig. 1d)[7]. Our data support a model whereby a thin back-skin epidermis at E12.5 mechanically favors a parallel division orientation in basal keratinocytes. As development proceeds and more suprabasal layers

are added, the thickened epidermis allows perpendicular cell division orientation in the basal layer. Then, in adult life, the skin epidermis is thin again and predominantly ensures parallel division orientation of basal keratinocytes. In our opinion, the correlations between a thinner epidermis and a parallel division orientation, or between a thicker epidermis and a perpendicular division orientation of the progenitors, can be explained by the topology of the epidermis restraining the cell division orientation, rather than the prevailing opposite causal relationship[10,61]. One example in this study is the *Sas-4* single mutants, which show a thinner epidermis and have a higher proportion of parallel divisions (Fig. 1f, Supplementary Fig. 3b).

In summary, our data support a two-phase model of physiological epidermal development (Fig. 7b). The first phase between E12.5–E15.5 is the main phase of stratification and is fueled by the higher proliferation rates of the basal cells and the newly-produced K10-positive and suprabasally-committed cells, which originate from the basal layer by cellular delamination (Fig. 7b). The second phase between E15.5–E18.5 is a maintenance phase, which is perhaps supported by cellular delamination from the basal layer and is contiguous with postnatal epidermal growth and turnover (Fig. 7b). Why and how certain basal progenitor cells commit to a cell fate switch, delamination, and differentiation in either phase are still open questions. For the first phase, cellular crowding and chance extrusion may be the dominant forces[55]. Perhaps the choice to delaminate is stochastic in nature, as has been shown for the adult skin epidermis and even other epithelia[54,59]. In this respect, a weak noisy signal, such as Notch signaling, is amplified and fixed by the cells committed to delaminate and differentiate, while concomitantly inhibiting the surrounding cells from adopting the same fate[6,41,62,63].

## Methods

**Animals and genotyping.** The following mouse alleles were used in this study: *Sas-4f/f* (Cenpj[tm1c(EUCOMM)Wtsi/tm1d(EUCOMM)Wtsi][23], *Ift88f/f* (Ift88[tm1Bky/tm1.1Bky][31], *K14-Cre*[34], *p53f/f* (Trp53[+tm1.Brn/tm1.1Brn][64], H2B-EGFP (CAG::H2B-EGFP)[65]. The CRISPR/Cas9 endonuclease-mediated knockout (em) mouse knockouts of *53bp1em2Baz* (*Trp53bp1em2Baz*) and *Usp28em2Baz* were generated by the CECAD in vivo Research Facility (ivRF) using microinjection of the corresponding gRNA (Supplementary Table 3), Cas9 mRNA, and Cas9 protein into fertilized zygotes.

All phenotypes were analyzed in the FVB/NRj background. The littermates that had the cre-recombinase and were heterozygous for the floxed or knockout alleles were used as preferred controls where available. Genotyping was carried out using standard PCR protocols and the primers listed in Supplementary Table 4.

The animals were generated, housed, and bred under standard conditions in the CECAD ivRF under a 12 h light cycle, at a temperature of $22 \pm 2\,°C$, $55 \pm 5\%$ relative humidity, and with food and water ad libitum. The generation and breeding described were approved by the Landesamt für Natur, Umwelt, und Verbraucherschutz Nordrhein-Westfalen (LANUV), Germany (animal applications: 84-02.04.2014.A372, 84-02.04.2015.A405 and 81-02.04.2019.A476).

**Human skin samples.** Human skin samples were obtained from different body sites with informed consent from both male and female individuals with an age range between 5 and 79 years. Paraffin sections of these samples from healthy individuals and individuals diagnosed with atopic dermatitis or psoriasis were provided by the Biobank of the SFB829 Z4 platform. The study protocol conformed to the ethical guidelines of the 1975 Declaration of Helsinki and was approved by the Ethics Committee of the Medical Faculty of the University of Cologne (Registration No. 12-163).

**Histological analysis.** Skin or embryo samples were fixed overnight in 10% Formalin or 4% paraformaldehyde (PFA), respectively, washed with 1× PBS, and stored in 70% ethanol for several days. After dehydration, the samples were embedded in paraffin and sectioned on an RM2255 microtome (Leica Biosystems) at 8 μm. For histological analyses, the skin sections were stained with hematoxylin and eosin (H&E) and mounted with Entellan® (Merck). The stained slides were imaged using a DM2000 light microscope (Leica biosystems) or scanned with an SCN400 Slide scanner (Leica biosystems) for subsequent analyses.

**Immunofluorescence and Imaging.** Embryos were fixed in 4% PFA overnight at 4 °C, washed with 1× PBS and then cryoprotected in 10–30% sucrose in 1× PBS overnight at 4 °C. After embedding in Tissue-Tek® (Optimal Cutting Temperature

Compound (OCT), Sakura Finetek USA INC), the blocks were sectioned on a CM1850 Cryostat (Leica Biosystems) at 7–10 μm. Postnatal skin samples for immunofluorescence staining were fresh-frozen in OCT and sectioned as above.

For immunofluorescence staining, skin sections were fixed for 10 min (min) in 4% PFA and washed with 1× PBS prior to the common staining protocol: skin or embryo sections were fixed in ice-cold methanol for 10 min at −20 °C, washed with washing buffer containing 0.2% Triton™ X-100 in 1× PBS and blocked for 1 h (h) in blocking buffer containing washing buffer and heat-inactivated goat serum (1% for embryo and 10% for skin sections). Mouse IgG Fab fragments were used at 1:10 to block background staining when using mouse primary antibodies (cat#115-007-003, Jackson Laboratories). After the primary antibodies were incubated overnight at 4 °C, the sections were washed with washing buffer, incubated with the secondary antibody and DAPI for 1 h at RT, and mounted with Prolong Gold (Cell Signaling). Images were obtained using an SP8 confocal microscope (Leica microsystems) or a Meta 710 confocal microscope (Zeiss) and image acquisition was performed using LAS X (Leica microsystems) or Zen lite (Zeiss) software, respectively.

For p63, p73, and p21 immunostaining, paraffin-embedded embryos were sectioned at 5 μm. Antigen retrieval of deparaffinized sections was performed using Retrieve-All buffer (BioLegend). Paraffin sections of human skin samples were deparaffinized and rehydrated before performing antigen retrieval by boiling in citrate buffer followed by the common staining protocol described above.

**Antibodies.** All primary and secondary antibodies and dyes used in this study are listed in Supplementary Table 5.

**Keratinocyte isolation.** Newborn mice were decapitated, transferred through a disinfection series (Betaisodona, 1× PBS 1:1; Octenisept; 1× PBS; 70% ethanol; 1× PBS; antibiotic/antimycotic solution in 1× PBS 1:100) and skinned. The skins were incubated overnight at 4 °C in a 5 mg/ml dispase II solution in DMEM/Hams-F12 (without supplements). After the epidermis was separated from the dermis, the epidermis was floated on the basal side down on 1 ml of 0.25% Trypsin (without EDTA, Gibco) for 20 min at room temperature (RT). Then, a keratinocyte medium was used to dissociate the keratinocytes from the epidermis[58,66].

**RNA isolation and RNA-Seq analyses.** RNA was prepared from frozen keratinocytes, which were isolated as described above without plating, or from frozen epidermal sheets of E13.5 embryos. Embryonic skin at E13.5 was micro-dissected and enzymatically separated into epidermis and dermis as described previously[67]. Briefly, the skin was incubated with 2:1 EDTA-free trypsin/pancreatin solution for 5 min at RT and 30 min on ice before separating the epidermis. Total RNA was isolated using the RNeasy Plus Mini Kit (Qiagen). RNA samples were submitted to the Cologne Center for Genomics (CCG), where quality checks and RNA-Seq were performed. The raw data were processed by the CECAD Bioinformatics facility using the QuickNGS pipeline to generate differentially expressed genes (fold change ≥2, p-value ≤ 0.05)[68]. For more information on the RNA-Seq procedure, please check the GEO deposited metadata file. Overlap of differentially expressed genes was determined and visualized in Venn diagrams using BioVenn[69]. Comparison of differentially expressed genes with canonical pathways gene sets derived from the PID pathway database[70] was analyzed using the Molecular Signatures Database v7.2 of GSEA[71].

**Skin barrier assay.** E17.5 or E18.5 embryos were sacrificed by making a cut in the neck to sever the spinal cord, and tail tips were taken for genotyping. After incubating the embryos for 2 min each in an increasing and then decreasing methanol series and washing in 1× PBS, they were stained in a 0.1% toluidine blue solution in water for 1–2 min on ice. A specific dye pattern showing possible barrier defects appeared after de-staining the embryos in 1× PBS on ice[7,72].

**Cell cycle analysis.** Freshly isolated epidermal keratinocytes were fixed in 70% ethanol and stored at −20 °C for several weeks. Then they were centrifuged at $77 \times g$, washed in 1× PBS and resuspended in propidium Iodide staining solution (10 μg/ml PI, 200 μg/ml RNAse A, 0.1% Triton™ X-100 in 1× PBS). After incubating at RT for 30 min, cell cycle analysis was performed using an LSRFortessa (BD) FACS machine. Data were analyzed using the FlowJo software based on the Watson pragmatic model (Supplementary Fig. 2d).

**Skin explant culture and EdU assays.** Lateral skin explants were taken from E13.5 wild-type embryos and placed on Nucleopore Track-Etch membranes (Whatman) floating on DMEM (Gibco) containing 10% FBS and 1× antibiotic/antimycotic solution (Gibco) at 37 °C and 5% $CO_2$. The explants were treated with either 10 μg/ml Mitomycin C or DMSO (vehicle) in the medium for 3 h, washed with 1× PBS, and cultured further in the medium above. Either directly at E13.5, after 1 day (E14.5) or after 2 days (E15.5), 20 μM EdU was added to the media for 2 h, then the explants were washed with 1× PBS and fixed in 4% PFA for 2 h at RT. After several washes with 1× PBS, the explants were removed from the membrane, cryoprotected in 30% sucrose overnight at 4 °C, and embedded in OCT for immunofluorescence staining (same protocol as for whole embryo sections except

using 0.1% Triton™ X-100 in PBS). For EdU analyses in embryonic mouse skin, the pregnant females at the corresponding time points were injected intraperitoneally with EdU at 50 mg/kg and then anesthetized and sacrificed 3 h later for embryo collection, fixation, and OCT embedding as described above. The skin sections were treated according to the Click-iT™ EdU imaging kit (Thermo Fisher) instructions before performing regular immunofluorescence staining.

**Time-lapse imaging**. Lateral skin explants were taken from E13.5 embryos (H2B-EGFP$^{tg/wt}$) and transferred to media (advanced DMEM + 2 mM L-glutamine + 0.1 mg/ml penicillin/streptomycin + 10% FBS) at 37 °C where they formed rolls. The rolls were embedded in 1% low-melting agarose or Matrigel (Corning) in media at 37 °C, where the cutting edge was touching the membrane-bottom of a Lumox® dish (Sarstedt) or a glass-bottom dish (ibidi). The dishes were covered with media and incubated at 37 °C and 5% $CO_2$ until imaging. Time-lapse imaging was carried out between E13.5 and E14.5 using an inverted SP8 confocal microscope (Leica microsystems), an inverted Dragonfly Spinning disc confocal microscope (Andor), or an inverted LSM710NLO Two-Photon microscope (Zeiss) with 20× air objective or 40× water or oil immersion objective and incubation at 37 °C and 5% $CO_2$.

**Image analyses**. Epidermal thickness and number of hair follicles were quantified by scanning the histological sections with the SCN400 slide scanner (Leica Biosystems) and analyzing the images using the ImageScope software (Leica Biosystems). The percentage of p53-positive cells in the epidermis was obtained using ImageJ (NIH) and CellProfiler (Broad Institute). The clusters of Cl.CASP3-positive cells, the thickness of K1 or K10 layers, the number and average size of basal and suprabasal layer cells, the Ki67-positive cells, the pHH3-positive cells as well as the centrosome- or cilia-containing cells in the epidermis were analyzed using ImageJ. The angles of basal cell division orientation were obtained from late anaphase or telophase cells in the epidermis by measuring the angle of the division axis (marked by Survivin) with the basement membrane (ITGA6) using ImageJ. The radial histograms were plotted using the OriginPro® software (Origin Lab), while all the other diagrams were generated using Prism (GraphPad). Time-lapse imaging data were manually analyzed using ImageJ (Correct 3D Drift plug-in), Volocity (Improvision), and Imaris (Bitplane).

**Statistics and reproducibility**. Two groups or more of data were compared using a two-tailed student's T-test with a cutoff for the significance of <0.05, or one-way ANOVA and Tukey's multiple comparisons test, both of which gave similar significance outcomes (Excel or GraphPad Prism). Angle measurements were compared using a two-way ANOVA, Kolmogorov–Smirnov Test and Chi-squared test (GraphPad Prism). The data are presented as the mean ± SD (standard deviation). Each experiment was repeated independently at least three times to ensure reproducibility.

**Supplementary theory**. In the following, we consider the basis of the two-phase model of embryonic epidermal development based on measurements of the net growth of the embryo, the change in the basal and suprabasal cell density, and estimates of the cell proliferation rate within the two layers. To determine the overall increase in epidermal basal and suprabasal cell numbers during development, we must combine estimates of the net increase in tissue area (as inferred from the net expansion of the embryo) and cell density. Both are measured with respect to anteroposterior (sagittal) and dorsoventral (transverse, hereafter called orthogonally) axes of the mouse (Fig. 6a; Supplementary Fig. 6b). Notably, the length of the embryo along both axes, $l_s(t)$ and $l_o(t)$, grows approximately linear with time over the entire E12.5–E18.5 time-course (Fig. 6a; Fig. S6b). Alongside this increase, there is also a change in the basal and suprabasal cell density, with the data showing a small differential between the sagittal and orthogonal directions, most likely due to the differential expansion rates along the body axes (Fig. 6b, c; Supplementary Fig. 6c, d). Therefore, if we define $\rho_{b/s,s/o}$ as the basal/suprabasal cell density in the sagittal/orthogonal directions, the total basal/suprabasal cell numbers increase in proportion to Eq. (1)

$$n_{b/s}(t) = \rho_{b/s,s}l_s \times \rho_{b/s,o}l_o$$

Based on this estimate, over the first three days from E12.5 to E15.5, the total cell numbers (basal and suprabasal) increase approximately exponentially with time, rising by a factor of around 7 (Fig. 6c; e; Supplementary Fig. 6d). Notably, this coincides with the time period in which proliferative cells are found in both basal and suprabasal cell layers (Fig. 6d; Supplementary Fig. 6e). After this period, the cell number increase in both basal and suprabasal cell layers is greatly reduced, showing only a factor of 2 increase over the next three days (Fig. 6c, e; Supplementary Fig. 6d).

Together, these results suggest a two-phase behavior, with an early phase of cell amplification through rounds of cell duplication, followed by a second phase in which basal cells steadily expand to meet the demands of the underlying growing tissue while, at the same time, giving rise to non-cycling suprabasal cells at a rate that allows both layers to expand in an approximately proportionate manner (Fig. 6c; Supplementary Fig. 6d). Significantly, during the early phase of cell

amplification, cells delaminate from the basal layer (potentially by the effects of cell crowding), giving rise to a proliferatively active suprabasal cell layer.

Therefore, to capture quantitatively the first phase of cell amplification, we suppose that the skin epidermis is comprised of equipotent progenitor cells, $p$, that duplicate through division at a constant rate $\lambda = 0.66$ per day according to Eq. (2):

$$p \xrightarrow{\lambda} p + p$$

Initially (E12.5–E13.5), this increase in cell number is accommodated through the ongoing expansion of the basal cell layer. During the next day (E13.5–E14.5), the basal cell density becomes elevated slightly, while an excess of proliferative cells is transferred—potentially through crowding—to the suprabasal cell layer. This trend continues during the third day of study (E14.5–E15.5), after which the ratio of basal to suprabasal cells reaches roughly parity (Fig. 6c; Supplementary Fig. 6d). At this time point, the frequency of proliferative suprabasal cells diminishes significantly (Fig. 6d; Supplementary Fig. 6e), suggesting that this last phase is characterized by cell cycle exit consistent with the cells' entry into a terminal differentiation program.

After this point, the data suggest only a modest rate of increase in the number of basal and suprabasal cells, which rise proportionately by a factor of around 2 over the next three days from E15.5 to E18.5 (Fig. 6c, e; Supplementary Fig. 6d). Although the experimental data is noisy, the measured increase has a linear-like trend. To model the dynamics in this phase (Fig. 6e), we suppose that the basal cell layer is comprised of a single equipotent progenitor cell population, $b$, and a single non-proliferative suprabasal cell population, $s$, defined by the kinetics[54], according to Eq. (3):

$$b \xrightarrow{\lambda} \begin{array}{ll} b + b & \text{Pr. } 1 - r \\ s & \text{Pr. } r \end{array}$$

so that at an overall rate $\lambda$, with probability $1 - r$, basal cells duplicate, while with probability $r$ they commit to terminal differentiation and stratify into the suprabasal layer, i.e., Eq. (4):

$$\dot{b} = \lambda(1 - 2r)b, \; \dot{s} = \lambda rb$$

To ensure that suprabasal cells are produced in proportion to basal cells, $\dot{b} = \dot{s}$, i.e., $r = 1/3$. Further, if we propose that, over the three-day time course, the proliferation rate, $\lambda$, is approximately constant, we have Eq. (5).

$$b = b_0 e^{\lambda(t-t_0)/3} = s$$

where $b_0$ denotes the basal cell number at time $t_0 = 15.5$ days. With basal cell number increasing by ~2 over the 3-day time course, this translates to a rate $\lambda = 0.7$ per day, similar to the estimated division rate during the early amplification phase.

As a consistency check, we can question what would be the expected EdU incorporation rate in the basal cell layer. With an S-phase of around $t_s = 6 - 8$ h, and a cell cycle time $1/\lambda$, we would expect a short-term EdU pulse to mark a fraction of $\lambda t_s \sim 0.8 \times (1/3) = 0.27$ of cells during the early phase. For a longer pulse (~3–4 h), marked cells would have progressed through one round of division, leading to a doubling of the number to around 50%. This figure is comparable to the measured estimates from the experimental data. In the later phase, the cell division rate drops by around 40%, broadly consistent with the EdU measurements (Fig. 6d).

**Reporting summary**. Further information on research design is available in the Nature Research Reporting Summary linked to this article.

## Data availability
The RNA-Seq data described in this study were deposited at the GEO repository under GSE161387. The PID pathway database used in this study is accessible via the Molecular Signatures Database v7.2 of GSEA (http://www.gsea-msigdb.org/gsea/msigdb/genesets.jsp?collection=CP:PID). All data supporting the findings of this study are available within the paper and its supplementary information and source data files, the mentioned repositories, or from the corresponding author upon reasonable request. Source data are provided with this paper.

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

## Acknowledgements

We thank our colleagues at the University of Cologne and SFB829: Carien Niessen, Sara Wickström (University of Helsinki, Finland), Sandra Iden (Saarland University, Germany), Catherin Niemann, Mirka Uhlirova, and Leo Kurian for critical discussions about the work and comments on the manuscript. We thank Terry Lechler (Duke University, USA) and Scott Williams (University of North Carolina, USA) for valuable suggestions during their sabbaticals in Cologne. We thank David Gonzalez and Valentina Greco (Yale University, USA) for their advice on time-lapse imaging analyses. We acknowledge and appreciate the CECAD in vivo research facility (Branko Zevnik) for the generation of the mouse lines of *53bp1 and Usp28*, and rederivation and maintenance of the other lines. We thank the CECAD imaging facility, especially Peter Zentis for image analyses, and Hans Fried and Christoph Möhl of the DZNE Imaging and Data Analysis Facilities (Bonn, Germany). Special thanks to Hironobu Fujiwara and Ritsuko Morita (RIKEN Center, Japan) for sharing unpublished data on vibrissae pad imaging. We acknowledge the SFB829 Z4 platform (Cornelia Mauch and Doris Helbig) for providing paraffin sections of normal and hyperproliferative human skin samples. The work was funded by the Deutsche Forschungsgemeinschaft (DFG, German Research Foundation)- Project-ID 73111208–SFB829 "Molecular Mechanisms regulating Skin Homeostasis", subproject A12 to H.B. The project was also supported by a seed funding to H.B. from the DFG under Germany's Excellence Strategy—CECAD, EXC 2030—390661388. B.D.S. acknowledges funding from the Wellcome Trust (219478/Z/19/Z) and the Royal Society in the form of an E.P. Abraham Research Professorship (RP\R1\180165). The funders had no role in study design, data collection, and analysis, decision to publish, or preparation of the manuscript.

## Author contributions

Conceptualization: H.B. and M.D.; Methodology: H.B., M.D., L.W. and B.D.S.; Software: E.S., M.D., L.W. and H.B.; Formal analysis: M.D., L.W., H.B., E.S. and B.D.S.; Investigation: M.D., L.W., E.S., H.K. and H.B.; Resources: C.K.; Writing: H.B., M.D. and L.W.; Visualization: M.D., L.W. and H.B.; Supervision, project administration, and funding acquisition: H.B.

## Funding

## Competing interests

The authors declare no competing interests.
