## [Peer Review File · Nature Communications]

REVIEWER COMMENTS

Reviewer #1 (Remarks to the Author):

The manuscript by Damen et al. entitled: "Epidermal stratification is uncoupled from centrosome dependent cell division orientation of the basal progenitors" investigates skin stratification during embryonic development in the absence of centrosomes. The authors find that even if spindle orientation defects are observed (similar to what has been reported in the original Sas4 fly mutant analysis) and in the mammalian brain by the last author, defects in epidermis stratification are not present. This major result challenges the current view that spindle orientation dictates or at least influences cell fate choices during mammalian embryonic skin stratification. This is a really important paper that deserves a lot of attention and it will be important for a large variety of researchers working in the brain, in the skin and in various epithelial tissues. In my opinion, the effort made by the authors is outstanding and the large majority of the data are extremely sound. However, I have a lot of comments because this work actually refutes and challenges past work that has been describing spindle orientation and asymmetric cell divisions as instrumental for cell fate. And somehow, the authors do not make this as a direct and more compelling message. This work deserves it and both the titles in the results, and the conclusions at the end of each paragraph should be more direct or straightforward. For such a big claim, which I am quite enthusiastic about it, the authors should focus on getting a revision that leaves no doubt to their claim. I am therefore really positive about this work and support its publication after revision. Also, for a non-skin specialist, some figures are difficult and one has to go to the legend or even to other papers to know what the markers are or what cell population we are looking at. Maybe in each figure having the identification of basal and suprabasal cells will help. And what the markers are labelling. Also, in Figure 1 a general model of the epidermis will help. Finally, the model even if beautiful is not at all informative. Including these changes plus the experiments mentioned above will make this article a really important paper for the community. Very well done!!!!

Major points

1) A more thorough and systematic analysis of the number of basal versus suprabasal cells (and not only layer thickness) in the same area should be included for all conditions analysed. This is essential to clarify the proportion of progenitor versus differentiated cells. For example, in Figures 3E-3F, the number of basal layer cells is increased in Sas4;p53KO skin, as compared to controls, but K1 layer thickness is not changed; does it mean that the number of suprabasal differentiated cells is not changed accordingly at this stage? Is there just a delay in differentiation? Including the number of cells in each category per area will certainly help clarifying this point.

2) The p53 null appears to generate defects in spindle orientation even if Sas4 is present- meaning independently of centrosome loss. How do the authors explain this?

3) The use of mouse model KO for USP28 and 53BP1 is a major plus in this article. Indeed, to my knowledge this is really the first time that animal models mutant for the mitotic surveillance pathway are being used. It would be nice if the authors could provide evidence of p53 activation status in this context. In principle, p53 should not be active, since these components of the mitotic surveillance are acting upstream but this should be included.

4) I am not a skin scientist so, I apologize if my question appears too dumb. I do not understand how

inhibiting proliferation, which in my opinion only causes a reduction in cell proliferation when using MMC allows to generate a stratified epidermis. What is K10 labelling? The authors mention cell delamination and that this might contribute to cell stratification. How is this possible? Maybe other explanations are required? I am also not sure of the importance of this experiment here specially in light of the importance of proliferation in the first stage for stratification.

5) In Figure 5, time-lapse analysis illustrates different behaviours of basal progenitor progeny and the possibility that cell delamination contributes to skin epithelium stratification. This is beautiful and highly relevant for this paper. However, as the results obtained, suggest a novel view supports the fact that spindle orientation is not related to cell fate, the authors should clearly identify the fate of basal and suprabasal cell progeny after live imaging of these cell divisions. In other words, they should include more direct lineage tracing experiments that identify the fate of cells being analysed. Maybe they can fixed and label different cell types after the time lapse? Also not sure how many movies were analysed and how many cells behaving in this way were noticed. This is really important to strongly support the author's conclusions.

6) The authors use a model based on parameters collected during embryonic development. Unfortunately, I am not capable of evaluating the model, but I guess that another reviewer is taking this task into his/her hands. They purpose the existence of two distinct steps that contribute to epidermis stratification. The first step takes place between E12.5 and E15.5 and cells undergo proliferation and causes stratification. In the second step or phase cells decrease proliferation and undergo terminal differentiation. While this appears as a plausible model with quite interesting implications, the authors have analysed most of their mutant conditions in what will be the second phase, where proliferation is less important. It is therefore essential to at least characterise cell divisions and spindle orientation in Ctrs and Sas4 mutants at these early stages.

7) After centrosome loss, p53 levels were high in most of the basal progenitor cells, while unexpectedly the number of apoptotic Caspase3-positive cells remained low (Fig2A-2C, FigS2A-2C), which indicates that skin thinning cannot be explained by apoptosis. They also exclude the possibility that G1 cell cycle arrest is responsible, but describe instead a delay in mitosis illustrated by an increased mitotic index (Figure S2E,2F). The authors propose in the discussion section as an explanation that p53-dependent decreased skin thickness results from a perturbation of p63 signalling. I think this is a farfetched option. It might be true, but without experimental data, bringing p63 to the equation appears irrelevant and actually not important for the overall message. However, what is happening to the longer mitosis is an important question. Maybe the time lapse movies can help clarifying this point. Even if they do not, this should be better discussed in the discussion section.

Minor points

1) For a non skin specialist the K6A, ITGa6, LOR etc.. are difficult to understand. For example the sentence: " the upregulation of K6A indicated an abnormal response in the Sas-4 mutant skin epithelium". What is K6A? What is the type of abnormal response? Maybe a scheme explaining the different layers and each individual marker would be adequate. Also, the quantification of each layer- in terms of cell size should also be included.

2) The title in the first subheading is some how not informative and maybe should include more information. In the first subheading, the authors show that the sas4 mice have: thinner epidermis,

- but the overall organization is maintained and that cilia are absolutely dispensable. Why is the subheading called: Centrioles are important for proper mouse epidermal and hair follicle development?. Maybe proper means something, but again it is too vague. And in my opinion, what they show is that centriole loss result in the surprising phenotype of not generating a total abnormal epidermis, and this is what it should be written, no? Even if hair loss is quite obvious- they conclude that “ despite the centriole deficiency in the basal keratinocyte progenitors, they showed robust regulation to allow the formation and maintenance of a generally functional skin barrier. “
- 3) The rescue of sas4 mutant with 53bp1 generates a mouse that appears smaller no (Figure 2E). Is there an explanation? Without considering the hair, it resembles the sas4 mut, while p53 and Usp28 rescue appear bigger.
 - 4) Page 9, the authors mention timepoints while wanting to refer to developmental stages- E16.5 and E17.5.
 - 5) The G is missing in Fig 3G and also H I guess.
 - 6) Maybe saying that MMC caused the complete inhibition of cell proliferation and cell division is a bit of an overstatement, no? Although largely reduced in terms of EdU and Ph3 numbers, I can still see some positive cells in Fig 4A. Maybe just mentioning that MCC highly reduced the number of Edu and Ph3 positive cells.
 - 7) Not sure I understand why the MMC incubation leads to such an increase (although not quantified) of Caspase 3 positive cells- Fig S4.

Reviewer #2 (Remarks to the Author):

The epidermis, the major barrier of our body, exhibits a multi-layered structure with sequential differentiation from basal to the horny layer. Developmentally, the epidermis first develops a simple or monolayered epithelium. At a certain time point in the embryonic stage, the epithelium starts to stratify. How the epidermis becomes stratified is an important question in both evolution and in development. P63 has been shown to be required for the transformation from a simple epithelium to a stratified epithelium in the epithelium. In the presence of P63, it is still not well understood how the epidermis transit from a monolayered structure to form suprabasal layers (including spinous layer to horny layer). One possibility is that basal progenitors preferentially undergo perpendicular division, or asymmetric division, to generate committed cells to form the suprabasal layer. In physiological adult mouse skin, it has been shown that stochastic delamination of single basal cells, but not perpendicular basal cell division, leads to the formation of suprabasal K10+ cells. The delamination of basal cells then triggers proliferation of adjacent basal cells to maintain an adequate basal cell density. In this work, authors try to test this by removing centrosome via knocking out Sas-4 in the epidermis in K14-Cre;Sas-4f/f. Their major findings include 1) when centrosome is absent, perpendicular cell division is increased from 40 to 60 % basal cell division, but the thickness of suprabasal K1+ cell layer is not significantly increased and basal cell density is slightly increased. 2) When cell division is suppressed by mitomycin treatment, basal cells can still delaminate to the suprabasal layer. 3) They characterize the cell dynamics of epidermis from E12.5 to E18.5 when both the body surface quickly expands and the epidermis becomes thickened into layered structures. They found that cell division in the basal and suprabasal layers is correlated with the fast increase of epidermal cell numbers in this period. The conclusion that basal delamination leads to formation of the suprabasal layer is largely consistent with the finding in the adult stage. As authors mentioned, epithelial stratification process begins before E12.5 and the first suprabasal layer generally appears

at E12.5. The experiments in this work mainly used embryonic skin from E13.5 and later stage. As authors showed (Figure 3, 4, 5), the epidermis already became stratified with at least some cells in the suprabasal layer at E13.5. Limited, at least in part, by this, the experiments and results might not be able to adequately dissect how perpendicular/parallel division contributes to the formation of the suprabasal layer (from simple epithelium to stratified epithelium). Experiments to use skin of earlier stages are needed. On the other hand, I think this work described and quantified the dynamics of how the epidermis progressively stratifies and expands during E12.5 to E18.5. I would suggest authors feature this. Authors showed division of suprabasal cells in this stage. Cell division of suprabasal cells in the physiological adult skin has not been clearly described in the literature. In contrast, division of suprabasal cells has been described in benign hyperplastic disease, such as psoriasis. Authors are suggested to discuss this. In addition, I have some other suggestions for authors.

Major points

1. In Figure 3D and Figure 5, authors analyzed the axis of cell division at E16.5 and E17.5. In E16.5 and E17.5, the epidermis is already stratified. Since the results from E16.5 and E17.5 reflect the cell division axis after epidermis has been stratified, they can not adequately address the issue that how cell division axis contributes to the transition from monolayer to stratified structures. Analysis of the axis of cell division during the transition from monolayer stage to the stratified stage should be more appropriate for the purpose. The authors primarily focused on the correlation between cell division orientation and epidermal stratification. I'll suggest authors demonstrate the stratification process prior to the morphological changes of SAS4 knockout. This will help the readers to understand this work better.
2. In K14cre; SAS4 f/f; p53 f/f and K14cre; SAS4 f/f (Fig.2A, 2E), the horny layer is compact but not basket-weave pattern shown in the controls. In K14cre; SAS4 f/f (Figure 2A), the epidermis is only thinner. At a closer look, the granular layer (and possibly the spinous layer) is also thinner. It seems that the differentiation process is altered when SAS4 is knocked out. I suggest authors to analyze further the epidermal changes, such as whether the spinous layer or the granular layer (how many cell layers in/thickness of spinous layer) is affected in the knockouts. If these are altered, it would be interesting to know the changes of differentiation related genes (or possibly Notch related genes) in transcriptional level. In Figure 2B, does the epidermal thickness include the horny layer?
3. In K14cre; SAS4 f/f, there are still intact hair preserved (Figure 2D). In the histological image, only some primitive hair germs were observed. Is the hair follicle development stuck in hair germ stage? Additionally, if centrosome is essential for normal hair follicle development/growth, is the preservation of intact hair follicles caused by incomplete knockout?
4. The authors established an animal model to investigate the function of SAS4 and the subsequent mitotic surveillance pathway during epidermal stratification. Authors need to describe the phenotypic change more clearly, such as quantifying the dynamics of the apoptosis and proliferation and including data of more time points. Since this study mainly focused on skin developmental stage during E13.5 to E18.5, I suggest authors provide more data during this period, instead of P0 or later stage. The histology of K14Cre; SAS4 f/f in E13.5 to E18.5 stage will help. In Figure S2F, it would be better to detect proliferating cells by pulse EdU. In Figure S2E, in order to determine the cell cycle in epidermal cells, authors should gate basal cells and suprabasal cells at least by integrin $\alpha 6$. In Figure 6, authors did not monitor the proliferation status in SAS4 depleted skin. The thinner epidermis would be the major phenotype in K14Cre; SAS4 f/f knockout skin. I suggest authors provide more information to demonstrate how SAS4 knockout affects epidermal morphogenesis.
5. During E12.5 to E18.5, there is fast expansion of the body surface and the tension is increased in

the skin. When skin is removed for explant culture, the tension is released. Therefore, the cell behavior in vitro (such as cell division) can be altered and might not reflect the condition in vivo. It has been shown that tension regulates cell behavior. Can authors consider and discuss this in the manuscript?

6. The author proposed a two-phase stratification model in the development of epidermis based on the quantification EdU and pHH3. Does the orientation of cell division differ in the two different phases?

Minor revision

1. In Figure 1, analysis of p63 expression in K14-Cre⁺; Sas-4f/f mice can help to interpret why the differentiation of stratified epitheliums can still occur.

2. In Figure 2A, it seems that p53 is mainly in the basal layer p53 and cleaved-caspase 3 is present in both the basal and suprabasal cells. Double staining with another basal cell marker, such as K14, K5, or integrin-alpha6, can help to delineate the local of p53 and cleaved-caspase 3. P53 expression is very extensive but only a few cells are positive for cleaved-caspase 3. Is there any explanation for this?

3. The authors utilized several genetic ablation mouse models, including K14cre; SAS4f/f, K14cre; SAS4 f/f; p53 f/f, K14cre; SAS4 f/f; 53bp1 f/f, and K14cre; SAS4 f/f; Usp28 f/f. In Figure3 B,C and Figure S3 A, they should also provide the cell division orientation of K14cre; SAS4f/f epidermal cells, even if they predicted the trend of cell division orientation might be the same in K14cre; SAS4 f/f; p53 f/f epidermal cells. I suggest authors provide more detailed data about the orientation of cell division in this research.

4. Authors used ITGA6 to mark basal cells or K10- suprabasal cells (Figure 3, 4, 5, 6) . Authors can consider staining for basement membrane (such as collagen IV). This will better show whether the cell body of cells with nucleus in the suprabasal position touches the basement membrane or they are truly suprabasal cells that already delaminate from the basal layer.

Reviewer #1:

The manuscript by Damen et al. entitled: "Epidermal stratification is uncoupled from centrosome dependent cell division orientation of the basal progenitors" investigates skin stratification during embryonic development in the absence of centrosomes. The authors find that even if spindle orientation defects are observed (similar to what has been reported in the original Sas4 fly mutant analysis) and in the mammalian brain by the last author, defects in epidermis stratification are not present. This major result challenges the current view that spindle orientation dictates or at least influences cell fate choices during mammalian embryonic skin stratification. This is a really important paper that deserves a lot of attention and it will be important for a large variety of researchers working in the brain, in the skin and in various epithelial tissues. In my opinion, the effort made by the authors is outstanding and the large majority of the data are extremely sound.

However, I have a lot of comments because this work actually refutes and challenges past work that has been describing spindle orientation and asymmetric cell divisions as instrumental for cell fate. And somehow, the authors do not make this as a direct and more compelling message. This work deserves it and both the titles in the results, and the conclusions at the end of each paragraph should be more direct or straightforward. For such a big claim, which I am quite enthusiastic about it, the authors should focus on getting a revision that leaves no doubt to their claim. I am therefore really positive about this work and support its publication after revision. Also, for a non-skin specialist, some figures are difficult and one has to go to the legend or even to other papers to know what the markers are or what cell population we are looking at. Maybe in each figure having the identification of basal and suprabasal cells will help. And what the markers are labelling. Also, in Figure 1 a general model of the epidermis will help. Finally, the model even if beautiful is not at all informative. Including these changes plus the experiments mentioned above will make this article a really important paper for the community. Very well done!!!!

We thank the reviewer for the positive evaluation and enthusiasm for our work. We are indeed excited that our data is supporting a new model in skin epidermal stratification that may apply as a more general paradigm in cell fate choices. However, given the substantial body of literature supporting the currently prevailing model and our ongoing experiments to further study the cell fate choices with new mouse models (see below), we are cautious with our data interpretation and present the direct conclusions while trying to avoid any leaps into speculation beyond the available data.

We have also provided a schematic of the skin epidermis in Fig. 1c to aid the readers in following our work and have modified the final model in Fig. 7b to better reflect our data interpretation.

Major points

1) A more thorough and systematic analysis of the number of basal versus suprabasal cells (and not only layer thickness) in the same area should be included for all conditions analysed. This is essential to clarify the proportion of progenitor versus differentiated cells. For example, in Figures 3E-3F, the number of basal layer cells is increased in *Sas4;p53KO* skin, as compared to controls, but K1 layer thickness is not changed; does it mean that the number of suprabasal differentiated cells is not changed accordingly at this stage? Is there just a delay in differentiation? Including the number of cells in each category per area will certainly help clarifying this point.

The number of basal and suprabasal cells as well as their average sizes are now provided in the new Fig. 4g-j and Fig. S4c,d and discussed in the text. The increase in basal cell density in *Sas-4 p53* double mutant epidermis is accompanied by an increase in the suprabasal cell density and both phenotypes seem to be dependent on p53 and IFT88/cilia (Fig. S4c). Of note, the change in cell division orientation in *Sas-4 p53* double mutant basal progenitors does not seem to be directly related to the changes in cell densities because the p53 mutant controls only show the latter.

2) The *p53* null appears to generate defects in spindle orientation even if *Sas4* is present- meaning independently of centrosome loss. How do the authors explain this?

Although there is trend for the *p53* mutant cells to slightly shift the cell division orientation compared to the controls, the differences are not statistically significant. From our analyses, we always find small differences in cell division orientations between genotypes and genetic backgrounds, all of which generate viable animals and intact epidermal barriers. In our opinion, these differences support our proposal that cell division orientation is not rigid, can be relatively flexible and may not directly reflect cell fate choices.

3) The use of mouse model KO for *USP28* and *53BP1* is a major plus in this article. Indeed, to my knowledge this is really the first time that animal models mutant for the mitotic surveillance pathway are being used. It would be nice if the authors could provide evidence of p53 activation status in this context. In principle, p53 should not be active, since these components of the mitotic surveillance are acting upstream but this should be included.

The p53 nuclear expression in the *Sas-4 53bp1* and *Sas-4 Usp28* mutants is now included and quantified (Fig. S3c,d). p53 nuclear levels are significantly different from *Sas-4* single mutants for *Sas-4 53bp1* but not for *Sas-4 Usp28*. As indicated in the text, the CRISPR/Cas9-generated mutations in *53bp1* or *Usp28* used in this study are both hypomorphic and not null alleles (Fig. S3a,b). Although the phenotypic rescues are clear and quantifiable (Fig. 3b-d), the data suggest that the hypomorphic mutations may affect the thresholds to p53 responses and, in this case, are not always reflected by the sensitivity of the p53 antibody.

4) I am not a skin scientist so, I apologize if my question appears too dumb. I do not understand how inhibiting proliferation, which in my opinion only causes a reduction in cell proliferation when using MMC allows to generate a stratified epidermis. What is K10 labelling? The authors mention cell delamination and that this might contribute to cell stratification. How is this possible.? Maybe other explanations are required? I am also not sure of the importance of this experiment here specially in light of the importance of proliferation in the first stage for stratification.

We agree with the astute comment from the reviewer because this point was not clear in the previous text. Our data showed that treating the skin cultures with MMC for 3 h at E13.5 was sufficient to **drastically** inhibit proliferation (both Ki67 and pHH3 in mitotic cells) after 24 h of culture (E14.5). This protocol is also commonly used to “mitotically-inactivate” fibroblast feeders (2 h of MMC). K10 marks the suprabasal differentiating keratinocytes, which are very few at E13.5 and form a continuous layer by E14.5 in control skin cultures. The K10-positive suprabasal cells were generated in the MMC-treated skin and were located on top of the basal cells after 24-48 h. Upon the inhibition of cell division with MMC, we concluded that K10-positive suprabasal cells **can** be generated after cell division inhibition, most likely by delamination, a process whose contribution to epidermal stratification and differentiation has been questioned and contested in the field.

We also agree that our new model emphasizes the high rate of cell division, including that of the K10-committed transit-amplifying cells, as the main driver of epidermal stratification. Although seemingly contradictory, our data from the MMC experiment suggested that cellular delamination might operate upon the inhibition of proliferation, but proliferation is still the main **physiological** driver of early stratification. In order to avoid this confusion and not over interpret the results of the MMC experiment, we have now moved it to supplementary data (Fig. S5a-e) and clarified the text merely to suggest the possible contribution of cellular delamination during epidermal differentiation.

5) In Figure 5, time-lapse analysis illustrates different behaviours of basal progenitor progeny and the possibility that cell delamination contributes to skin epithelium stratification. This is beautiful and highly relevant for this paper. However, as the results obtained, suggest a novel view supports the fact that spindle orientation is not related to cell fate, the authors should clearly identify the fate of basal and suprabasal cell progeny after live imaging of these cell divisions. In other words, they should include more direct lineage tracing experiments that identify the fate of cells being analysed. Maybe they can fixed and label different cell types after the time lapse? Also not sure how many movies were analysed and how many cells behaving in this way were noticed. This is really important to strongly support the author's conclusions.

We agree with the reviewer that a cell fate reporter would be required to make a definitive conclusion on cell behavior and exchange between the epidermal layers. We are currently generating mouse models with reporters on basal and suprabasal cell fates for future experiments. Our current data are suggestive and provide a new model for stratification that requires further experimental testing with the new mouse models. Although we are quite excited about our findings, we remain cautious on data interpretation. The experiments in fixed tissues suggested by the reviewer are currently not feasible in our system to trace individual cells without live reporters. The number of movies analyzed and cells observed are mentioned in the figure legend (Fig. 5, 5 movies from 5 independent imaging experiments, number of cells: a = 44 out of 94 cells, b = 50 out of 94 cells, c = 46 cells).

6) The authors use a model based on parameters collected during embryonic development. Unfortunately, I am not capable of evaluating the model, but I guess that another reviewer is taking this task into his/her hands. They propose the existence of two distinct steps that contribute to epidermis stratification. The first step takes place between E12.5 and E15.5 and cells undergo proliferation and causes stratification. In the second step or phase cells decrease proliferation and undergo terminal differentiation. While this appears as a plausible model with quite interesting implications, the authors have analysed most of their mutant conditions in what will be the second phase, where proliferation is less important. It is therefore essential to at least characterise cell divisions and spindle orientation in Ctrs and Sas4 mutants at these early stages.

We agree with the reviewer that our new model highlights the importance of the first phase in epidermal stratification. We analyzed cell division orientation at E16.5 and E17.5 because the biphasic distribution of cell division angles seems to be established at these timepoints (Williams et al, 2011). Earlier division angles at E14-E15 seem more random. In fact, there is no consensus in the field about which **embryonic stage** to consider, and also which phase of mitosis to analyze, for example **metaphase versus anaphase/telophase** (when the final orientation is set). We have now discussed these issues and included an earlier timepoint, E15.5, which showed a more random distribution and no difference between controls and Sas-4 p53 double mutants. In our opinion, the relationship between cell division orientation and cell fate, if it plays a role, is most likely conserved across stages.

7) After centrosome loss, p53 levels were high in most of the basal progenitor cells, while unexpectedly the number of apoptotic Caspase3-positive cells remained low (Fig2A-2C, FigS2A-2C), which indicates that skin thinning cannot be explained by apoptosis. They also exclude the possibility that G1 cell cycle arrest is responsible, but describe instead a delay in mitosis illustrated by an increased mitotic index (Figure S2E,2F). The authors propose in the discussion section as an explanation that p53-dependent decreased skin thickness results from a perturbation of p63 signalling. I think this is a farfetched option. It might be true, but without experimental data, bringing p63 to the equation appears irrelevant and actually not important for the overall message. However, what is happening to the longer mitosis is an important question. Maybe the time lapse movies can help clarifying this point. Even if they do not, this should be better discussed in the discussion section.

As the reviewer mentioned, a fraction of p53-positive cells that perhaps reaches a certain threshold, or p53 oscillation dynamics (Purvis JE,..., Lahav G, *Science* 2012), undergoes cell death during epidermal development before P0 (new Fig. 2a-e) and this can at least contribute to a resulting thinner epidermis. p63 is the master regulator of epidermal stratification at the earliest level of commitment to an epidermal fate. p63 and p53 share similar, if not identical, DNA-binding motifs. In the WT epidermis, p53 is barely detectable, while in *Sas-4* mutant keratinocytes p53 is ectopically high. We have added new experiments with RNA-Seq data showing that p53-regulated genes significantly overlap with p63-bound gene-regulatory elements (Fig. 2h-j). The data suggest that p53 might displace p63 at the DNA elements of its regulated genes and thereby disrupt epidermal homeostasis. Data in cultured mammalian cells have shown that the longer mitotic duration in the absence of centrioles results in an increase in nuclear p53 following mitosis and we believe a similar mechanism operates in the epidermis.

Minor points

1) *For a non skin specialist the K6A, ITGa6, LOR etc.. are difficult to understand. For example the sentence: "the upregulation of K6A indicated an abnormal response in the Sas-4 mutant skin epithelium". What is K6A? What is the type of abnormal response? Maybe a scheme explaining the different layers and each individual marker would be adequate. Also, the quantification of each layer- in terms of cell size should also be included.*

K6A is expressed in activated keratinocytes during stress or in skin disease conditions. We have added the data showing that K6A expression in *Sas-4* mutants is dependent on p53 (Fig. 3e) and explained it better in the text. The scheme of the epidermis has been added in (Fig. 1c). The quantifications of the cell numbers and sizes in the basal and suprabasal layers for Ctrl, p53, *Sas4 p53*; and *ft88* have been added (Fig. 4g-j and Fig. S4c,d).

2) *The title in the first subheading is some how not informative and maybe should include more information. In the first subheading, the authors show that the sas4 mice have: thinner epidermis, but the overall organization is maintained and that cilia are absolutely dispensable. Why is the subheading called: Centrioles are important for proper mouse epidermal and hair follicle development?. Maybe proper means something, but again it is too vague. And in my opinion, what they show is that centriole loss result in the surprising phenotype of not generating a total abnormal epidermis, and this is what it should be written, no? Even if hair loss is quite obvious- they conclude that " despite the centriole deficiency in the basal keratinocyte progenitors, they showed robust regulation to allow the formation and maintenance of a generally functional skin barrier. "*

We have changed the heading to "Centrioles are important, but not essential, for skin epidermal and hair follicle development" to convey both messages.

3) *The rescue of sas4 mutant with 53bp1 generates a mouse that appears smaller no (Figure 2E). Is there an explanation? Without considering the hair, it resembles the sas4 mut, while p53 and Usp28 rescue appear bigger.*

We have corrected the scaling in the different mouse pictures. Although there are minor differences in animal sizes at this stage, there is no consistent difference in any particular genotype except for the *Sas-4* mutants, which are smaller than the other genotypes (Fig. S1a).

4) *Page 9, the authors mention timepoints while wanting to refer to developmental stages- E16.5 and E17.5.*

The wording has been corrected to "stages".

5) *The G is missing in Fig 3G and also H I guess.*

The previous panels in Fig. 3 next to E and F belonged to B and C, respectively. This figure has changed and the new labels are better arranged.

6) *Maybe saying that MMC caused the complete inhibition of cell proliferation and cell division is a bit of an overstatement, no? Although largely reduced in terms of EdU and Ph3 numbers, I can still see some positive cells in Fig 4A. Maybe just mentioning that MCC highly reduced the number of Edu and Ph3 positive cells.*

As discussed above for the MMC-treated skin, we observed some pHH3 in cells in interphase, perhaps G2 phase, but not mitotic cells, which typically show condensed chromosomes and intense staining. For Ki67, it is close to background levels. This is now discussed in the text and the figure has been moved to supplementary data (Fig. S5a-e).

7) *Not sure I understand why the MMC incubation leads to such an increase (although not quantified) of Caspase 3 positive cells- Fig S4.*

MMC treatment leads to replication stress-induced DNA damage, which in a developing tissue like the skin causes p53 upregulation and cell death (Cl.CASP3). We understand the limitations of this experiment and only used it to show that when proliferation and cell division are inhibited, the basal keratinocytes can still generate suprabasal cells and move upwards, most likely by delamination.

Reviewer #2:

The epidermis, the major barrier of our body, exhibits a multi-layered structure with sequential differentiation from basal to the horny layer. Developmentally, the epidermis first develops a simple or monolayered epithelium. At a certain time point in the embryonic stage, the epithelium starts to stratify. How the epidermis becomes stratified is an important question in both evolution and in development. P63 has been shown to be required for the transformation from a simple epithelium to a stratified epithelium in the epithelium. In the presence of P63, it is still not well understood how the epidermis transit from a monolayered structure to form suprabasal layers (including spinous layer to horny layer). One possibility is that basal progenitors preferentially undergo perpendicular division, or asymmetric division, to generate committed cells to form the suprabasal layer. In physiological adult mouse skin, it has been shown that stochastic delamination of single basal cells, but not perpendicular basal cell division, leads to the formation of suprabasal K10+ cells. The delamination of basal cells then triggers proliferation of adjacent basal cells to maintain an adequate basal cell density. In this work, authors try to test this by removing centrosome via knocking out Sas-4 in the epidermis in K14-Cre;Sas-4^{fl/fl}. Their major findings include 1) when centrosome is absent, perpendicular cell division is increased from 40 to 60 % basal cell division, but the thickness of suprabasal K1+ cell layer is not significantly increased and basal cell density is slightly increased. 2) When cell division is suppressed by mitomycin treatment, basal cells can still delaminate to the suprabasal layer. 3) They characterize the cell dynamics of epidermis from E12.5 to E18.5 when both the body surface quickly expands and the epidermis becomes thickened into layered structures. They found that cell division in the basal and suprabasal layers is correlated with the fast increase of epidermal cell numbers in this period. The conclusion that basal delamination leads to formation of the suprabasal layer is largely consistent with the finding in the adult stage. As authors mentioned, epithelial stratification process begins before E12.5 and the first suprabasal layer generally appears at E12.5. The experiments in this work mainly used embryonic skin from E13.5 and later stage. As authors showed (Figure 3, 4, 5), the epidermis already became stratified with a least some cells in the suprabasal layer at E13.5. Limited, at least in part, by this, the experiments and results might not be able to adequately dissect how perpendicular/parallel division contributes to the formation of the suprabasal layer (from simple epithelium to stratified epithelium). Experiments to use skin of earlier stages are needed. On the other hand, I think this work described and quantified the dynamics of how the epidermis progressively stratifies and expands during E12.5 to E18.5. I would suggest authors feature this. Authors showed division of suprabasal cells in this stage. Cell division of suprabasal cells in the physiological adult skin has not been clearly described in the literature. In contrast, division of suprabasal cells has been described in benign hyperplastic disease, such as psoriasis. Authors are suggested to discuss this. In addition, I have some other suggestions for authors.

We thank the reviewer for the expert summary of the literature and the insightful comments on the essential contribution of the early stages of epidermal stratification, which is in line with our model and final conclusions. Although our views completely coincide with the reviewer's, most of the published literature on epidermal stratification and progenitor division orientation is largely focused on the later stages (E16.5-E17.5), which show a biphasic distribution of parallel and perpendicular divisions (Williams et al, 2011). Earlier stages (E14.5-E15.5) show a more random distribution of the cell division angles, while the initiating stages (E9.5-E13.5) show predominantly parallel divisions. In addition, the analyses are also dependent on the phase of mitosis, metaphase versus anaphase/telophase, which adds to the variability in the published literature. We have analyzed earlier

stages in the current version, emphasized our new model and added data pertaining to human skin disease (Fig. 7a).

Major points

1. In Figure 3D and Figure 5, authors analyzed the axis of cell division at E16.5 and E17.5. In E16.5 and E17.5, the epidermis is already stratified. Since the results from E16.5 and E17.5 reflex the cell division axis after epidermis has been stratified, they can not adequately address the issue that how cell division axis contributes to the transition from monolayer to stratified structures. Analysis of the axis of cell division during the transition from monolayer stage to the stratified stage should be more appropriate for the purpose. The authors primarily focused on the correlation between cell division orientation and epidermal stratification. I'll suggest authors demonstrate the stratification process prior to the morphological changes of SAS4 knockout. This will help the readers to understand this work better.

As discussed above, we focused on the later stages because they are more relevant for the published literature. We have now analyzed earlier stages and added the data from the E15.5 (Fig. 4b). The data showed that the distribution of cell division angles is not significantly different between controls and *Sas-4 p53* double mutants, and followed a random distribution among the three bins without preferential bias to parallel or perpendicular divisions (two-way ANOVA and Kolmogorov-Smirnov test). In addition, in the figure below, we provide the data from the E12.5 stage, which also showed no difference. We also note that the phenotype in *Sas-4* mutants is significant and evident from E14.5 onwards (new Fig. 2a). In our opinion, the data from the time-lapse imaging (Fig. 5 and Movies S1-S3) and mathematical calculations of growth, cell numbers and proliferation (Fig. 6 and S6), support our new model of how a stratified epidermis can be generated, mainly by proliferation of the progenitors and transit-amplifying cells. For the exact relationship between cell division and cell fate in both cell populations, new mouse reporter lines are required, which we are in the process of generating and will be the focus of future studies.

2. In *K14cre; SAS4 f/f; p53 f/f* and *K14cre; SAS4 f/f* (Fig.2A, 2E), the horny layer is compact but not basket-weave pattern shown in the controls. In *K14cre; SAS4 f/f* (Figure 2A), the epidermis is only thinner. At a closer look, the granular layer (and possibly the spinous layer) is also thinner. It seems that the differentiation process is altered when *SAS4* is knocked out. I suggest authors to analyze further the epidermal changes, such as whether the spinous layer or the granular layer (how many cell layers in/thickness of spinous layer) is affected in the knockouts. If these are altered, it would be interesting to know the changes of differentiation related genes (or possibly Notch related genes) in transcriptional level. In Figure 2B, does the epidermal thickness include the horny layer?

The data support the conclusion that the differentiation process is altered in the *Sas-4* single mutants. The epidermal thickness, excluding the cornified layer, was thinner in these mutants, and the individual differentiation markers also appeared thinner. The basket weave cornified layer was also consistently affected in histology from the single mutants at P0 (Fig. 1e) but not at P8, for example (Fig. S1c). The cornified layer did not always appear as compact for *Sas-4 p53* double mutants (Figure on the right).

To assess the transcriptional changes in *Sas-4* single mutants (at E13.5 and P0), we performed RNA-Seq experiments (Fig. 2h-j and Tables S1, S2). The data have been added and discussed in the text. In brief, the data showed, as expected, that p53 downstream signaling and p63/p73 pathways were the predominantly enriched pathways in the *Sas-4* single mutants at both E13.5 and P0. The long list of differentially expressed genes in the *Sas-4* single mutant keratinocytes (over 3600 genes) indicated highly disturbed differentiation programs and perhaps compensatory mechanisms.

3. In *K14cre; SAS4 f/f*, there are still intact hair preserved (Figure 2D). In the histological image, only some primitive hair germs were observed. Is the hair follicle development stuck in hair germ stage? Additionally, if centrosome is essential for normal hair follicle development/growth, is the preservation of intact hair follicles caused by incomplete knockout?

Some hair follicles can still form in *Sas-4* single mutants and are indeed delayed in their development by comparing the histology at P0, P8 and P21 (Fig. 1e, Fig. S1b,c). The *K14-Cre* used is efficient, acts early ~E9.5 (Hafner et al, 2004) and the number of centrosomes is extremely low even at E15.5 (Fig. 1b). In fact, when the same knockout is performed on a C57BL/6 background, the mouse is almost completely bald, except for the vibrissae hairs (Figure on the right). The FVB/N background phenotype reported in our study is milder perhaps due to modifier genes and differential p53 thresholds.

4. The authors established an animal model to investigate the function of *SAS4* and the subsequent mitotic surveillance pathway during epidermal stratification. Authors need to describe the phenotypic change more clearly, such as quantifying the dynamics of the apoptosis and proliferation and including data of more time points. Since this study mainly focused on skin developmental stage during E13.5 to E18.5, I suggest authors provide more data during this period, instead of P0 or later stage. The histology of *K14Cre; SAS4 f/f* in E13.5 to E18.5 stage will help. In Figure S2F, it would be better to detect proliferating cells by pulse EdU. In Figure S2E, in order to determine the cell cycle in epidermal cells, authors should gate basal cells and suprabasal cells at least by integrin $\alpha 6$. In Figure 6, authors did not monitor the proliferation status in *SAS4* depleted skin. The thinner epidermis would be the major phenotype in *K14Cre; SAS4 f/f* knockout skin. I suggest authors provide more information to demonstrate how *SAS4* knockout affects epidermal morphogenesis.

In this work, we did not focus on the *Sas-4* single mutant phenotype because the outcome of p53-dependent abnormalities was rather expected from our previous publications on the developing embryo and developing brain (Bazzi et al, PNAS 2014; Insolera, Bazzi et al, Nature Neurosci, 2014). Our aim was to use the *Sas-4 p53* double mutants, which bypass the pathway to disrupt cell division orientation (Insolera, Bazzi, 2014). Upon the request of this reviewer, we have added a new time-course of immunostaining with TUBG and p53, quantified p53-positive cells earlier, and combined it with the previous data from E15.5 and P0 (new Fig. 2a,b). The RNA-Seq data on epidermal cells at E13.5 and P0 also provide more information on the *Sas-4* single mutant phenotype (Fig. 2h-j, Table S1, S2). The activation of the mitotic surveillance pathway in mammalian cells *in vitro* causes G1 cell cycle arrest and that is why we used cell cycle analyses using propidium iodide and FACS (Fig. 2f), which ruled out an arrest in G1 even in the presence of high p53. We used EdU pulse experiments for the other genotypes (Fig. S4e). We have also stained the freshly isolated primary mouse epidermal keratinocytes, similar to the ones used for the cell cycle experiments, with ITGA6. A similar proportion of basal keratinocytes is present in control and *Sas-4* single mutants (~80% basal and ~20% suprabasal, with n=2 each).

5. During E12.5 to E18.5, there is fast expansion of the body surface and the tension is increased in the skin. When skin is removed for explant culture, the tension is released. Therefore, the cell behavior *in vitro* (such as cell division) can be altered and might not reflect the condition *in vivo*. It has been shown that tension regulates cell behavior. Can authors consider and discuss this in the manuscript?

We thank the reviewer for this comment and suggestion. We have added this point to our discussion. Although the skin explant cultures are released from tension because they are removed from the embryo, as we point out in our manuscript, they still stratify, differentiate and even form well-patterned hair follicles, as has been shown by many publications including ours (Fig. S5h, Richardson, Bazzi et al, Development 2009). In addition, our new skin roll cultures that are embedded in agarose or matrigel (see Methods), better recapitulate the epidermal stratification and differentiation program *in vivo* (Fig. S5g), perhaps due to the pressure offered by the embedding matrix. In our opinion, the ideal conditions for imaging the developing skin *in utero* have not been established yet, and even imaging the developing skin while still on the embryo, as has been recently published (For example, Ouspenskaia, 2016 and Miroshnikova, 2018), has to be taken with the caveat that the embryo is already dead and perhaps not enough nutrients are reaching the skin from the inside.

6. The author proposed a two-phase stratification model in the development of epidermis based on the quantification EdU and pHH3. Does the orientation of cell division differ in the two different phases?

Please see the response above for point #1.

Minor revision

1. In Figure 1, analysis of p63 expression in K14-Cre⁺; Sas-4^{f/f} mice can help to interpret why the differentiation of stratified epitheliums can still occur.

The staining for p63 and p73 have now been added (Fig. S2c) and discussed in the text in the context of the new RNA-Seq data.

2. In Figure 2A, it seems that p53 is mainly in the basal layer p53 and cleaved-caspase 3 is present in both the basal and suprabasal cells. Double staining with another basal cell marker, such as K14, K5, or integrin-alpha6, can help to delineate the local of p53 and cleaved-caspase 3. P53 expression is very extensive but only a few cells are positive for cleaved-caspase 3. Is there any explanation for this?

We agree with the reviewer's assessment and as suggested performed immunostaining for ITGA6 together with either p53 or Cl. CASP3 at E15.5 (Figure on the right). In our opinion, a fraction of p53-positive cells that perhaps reaches a certain threshold, or p53 oscillation dynamics (Purvis JE,..., Lahav G, Science 2012), undergoes cell death during epidermal development before P0.

3. The authors utilized several genetic ablation mouse models, including K14cre; SAS4^{f/f}, K14cre; SAS4 ^{f/f}; p53 ^{f/f}, K14cre; SAS4 ^{f/f}; 53bp1 ^{f/f}, and K14cre; SAS4 ^{f/f}; Usp28 ^{f/f}. In Figure3 B,C and Figure S3 A, they should also provide the cell division orientation of K14cre; SAS4^{f/f} epidermal cells, even if they predicted the trend of cell division orientation might be the same in K14cre; SAS4 ^{f/f}; p53 ^{f/f} epidermal cells. I suggest authors provide more detailed data about the orientation of cell division in this research.

We have added the cell division orientation of the Sas-4 single mutants (E16.5, Fig. S4b). The data suggested that the thinner epidermis in these mutants is associated with an increase in the proportion of parallel division orientations and a decrease in perpendicular ones, which is consistent with our model that a thinner epidermis topologically favors parallel division orientations (see Discussion).

4. Authors used ITGA6 to mark basal cells or K10- suprabasal cells (Figure 3, 4, 5, 6). Authors can consider staining for basement membrane (such as collagen IV). This will better show whether the cell body of cells with nucleus in the suprabasal position touches the basement membrane or they are truly suprabasal cells that already delaminate from the basal layer.

As suggested by the reviewer, we have added immunostainings for COL4A4 (E16.5, Fig. S5i) as well as LAMA1 (Fig. S5j). Both were localized at the basement membrane and not surrounding the cells like ITGA6. To assess whether a cellular process projects from a cell in the "second" layer and touches

the basement membrane requires a careful chimeric analysis of individually labelled cellular cytoplasm or membrane. In our opinion, a more definitive method to assess cell fate is by using fate reporters, which is our aim for future studies.

We hope that you find the new data and analyses sufficient to support our conclusions and merit publication in Nature Communications.

Kind regards,
Hisham Bazzi

REVIEWERS' COMMENTS

Reviewer #1 (Remarks to the Author):

The authors have addressed all my comments and therefore I think this manuscript is ready to be published.

I am not sure about the results concerning the RNAseq in the centriole mutant background. To me, this is almost like a different question and it seems displaced from the main message of the paper.

What I still think that deserves attention relates with the fate of the centriole less cells, even if it does not require more experiments. The authors see p53 up regulation and more time in mitosis. They argue that this was already known. I agree, but in the various studies published so far, mammalian cells without centrosomes that spend too much time in mitosis die of apoptosis or arrest in G1. If I understand things correctly, this is not the case here, yes? Maybe this just has to be stated.

The title is not very informative, maybe this can be changed?

Congratulations on this study, it will be extremely important to the field.

Reviewer #3 (Remarks to the Author):

Authors performed comprehensive revisions, that address all major criticism.